# Nationwide health, socio-economic and genetic predictors of COVID-19 vaccination status in Finland

Tuomo Hartonen [1,10], Bradley Jermy [1,10], Hanna Sõnajalg[2], Pekka Vartiainen [1], Kristi Krebs [2], Andrius Vabalas [1], FinnGen*, Estonian Biobank Research Team*, Tuija Leino[3], Hanna Nohynek[3], Jonas Sivelä[3], Reedik Mägi[2], Mark Daly[1,4,5,6], Hanna M. Ollila[1,4,7,8], Lili Milani [2], Markus Perola[3], Samuli Ripatti[1,4,5,9] & Andrea Ganna [1,4,5] ✉

Understanding factors associated with COVID-19 vaccination can highlight issues in public health systems. Using machine learning, we considered the effects of 2,890 health, socio-economic and demographic factors in the entire Finnish population aged 30–80 and genome-wide information from 273,765 individuals. The strongest predictors of vaccination status were labour income and medication purchase history. Mental health conditions and having unvaccinated first-degree relatives were associated with reduced vaccination. A prediction model combining all predictors achieved good discrimination (area under the receiver operating characteristic curve, 0.801; 95% confidence interval, 0.799–0.803). The 1% of individuals with the highest predicted risk of not vaccinating had an observed vaccination rate of 18.8%, compared with 90.3% in the study population. We identified eight genetic loci associated with vaccination uptake and derived a polygenic score, which was a weak predictor in an independent subset. Our results suggest that individuals at higher risk of suffering the worst consequences of COVID-19 are also less likely to vaccinate.

In the face of the COVID-19 pandemic, several laboratories developed safe and effective vaccines in record-breaking time[1]. However, across high-income countries, somewhere between 5% and 30% of the population has not received a single dose of a COVID-19 vaccine. Even higher proportions of populations in low-income countries remain unvaccinated[2]. In Finland, 23.5% of the population had not received a single dose of a COVID-19 vaccine by the end of October 2021. A broad and rapid vaccination against COVID-19 helps reduce disease severity (vaccination effectiveness against death, 99.0%[3]), the health-care burden (vaccination effectiveness against hospitalization, 97.2%[3]) and the spread of infection[4]. Refusal, postponement or inability to participate in the vaccination programme is therefore a key societal concern. Being able to identify individual factors impacting vaccination uptake can help policymakers design more effective targeted interventions for future immunization programmes.

[1]Institute for Molecular Medicine Finland, FIMM, HiLIFE, University of Helsinki, Helsinki, Finland. [2]Estonian Genome Centre, Institute of Genomics, University of Tartu, Tartu, Estonia. [3]Finnish Institute for Health and Welfare, Helsinki, Finland. [4]Broad Institute of MIT and Harvard, Cambridge, MA, USA. [5]Massachusetts General Hospital, Cambridge, MA, USA. [6]Harvard Medical School, Cambridge, MA, USA. [7]Center of Genomic Medicine, Harvard Medical School, Boston, MA, USA. [8]Anesthesia, Critical Care, and Pain Medicine, Massachusetts General Hospital and Harvard Medical School, Boston, MA, USA. [9]Department of Public Health, University of Helsinki, Helsinki, Finland. [10]These authors contributed equally: Tuomo Hartonen, Bradley Jermy. *Lists of authors and their affiliations appear at the end of the paper. ✉e-mail: andrea.ganna@helsinki.fi

Several previous studies on the correlates of COVID-19 vaccination were based on surveys[5–11]. They have found that trust and knowledge about the vaccine, recommendations by health-care professionals, beliefs about the severity of the disease, and convenience of vaccination were important correlates of vaccination intentions. This is in line with previous studies about vaccine hesitancy[12,13].

Nevertheless, studies based on survey data have limitations. First, surveys usually include only a few thousand individuals, and this limits their statistical power. Second, populations included in surveys are often not representative of the general population, and factors associated with vaccine hesitancy (such as socio-economic status or education level) are also associated with participation in scientific studies[14]. People less likely to get a vaccine are more likely to be under-represented in these studies. Third, surveys include only a limited set of information, limiting the power of epidemiological and machine learning analyses.

Here we used a comprehensive collection of nationwide registers covering detailed health, socio-economic, familial and demographic information to map potential predictors of COVID-19 vaccination uptake across the entire Finnish population (5.5 million individuals). We compared 2,890 predictors measured before 31 December 2019 and the uptake of the first dose of a COVID-19 vaccine between 27 December 2020 and 31 October 2021. We used machine learning methods to quantify the importance of 12 predictor categories (such as disease history, medication purchases and education level; Fig. 1a) and their overlap. Finally, we combined these categories to derive a prediction model of COVID-19 vaccination status.

Previous studies have shown a genetic liability and identified individual genetic factors that impact COVID-19 severity and susceptibility[15]. Across 273,765 individuals (with replication in an additional 145,615 individuals from Estonia), we evaluated whether genetic information could predict COVID-19 vaccination uptake, whether there is a genetic overlap with health and behavioural traits that were not available nationwide, and whether individuals with higher genetic risk for COVID-19 were more or less likely to be vaccinated. This study establishes a framework for using machine learning and statistical genetics methods to identify individuals that are less likely to participate in COVID-19 vaccination programmes.

## Results

### Nationwide data to identify predictors of COVID-19 vaccination

The FinRegistry project (https://www.finregistry.fi/) combines and harmonizes data from 18 Finnish nationwide registers into a comprehensive dataset for epidemiological and machine learning analyses. Briefly, these registers cover disease diagnoses from primary, secondary and tertiary care; medication purchases; welfare benefits; multi-generational familial relationships; and socio-economic and demographic information for at least ten years, with some registers dating back to the 1970s (Fig. 1 and Methods). One of these registers, the Finnish Vaccination Register, contains records of all COVID-19 vaccination doses administered in Finland.

We manually divided the data, 2,890 potential predictors in total, into 12 consistent categories for easier interpretation of the results. Predictors were available before 31 December 2019 (that is, before the start of the COVID-19 pandemic, except for the vaccination status of relatives, for which vaccination records until 31 October 2021 were used) for all individual residents of Finland alive on 31 December 2020. We considered only individuals between 30 and 80 years old and excluded 6.1% of the study population who had emigrated and a further 1.9% with a reported positive COVID-19 test by 31 October 2021. We further excluded 0.1% of the remaining study population living in Askola, a municipality with incomplete vaccination records (Extended Data Fig. 1). We chose the age range 30–80 because by 31 October 2021 everyone in this age range had been eligible for a first dose of COVID-19

for at least four months (Fig. 1c). In total, we included 3,192,505 individuals (50.5% females), of which 136,947 women (8.5%) and 171,647 men (10.9%) (Fig. 1b) were unvaccinated. Younger individuals were eligible for vaccination later and had a lower vaccination rate by the end of the study period (Fig. 1c). Age was thus used as a covariate in all the presented analyses. Genetic information from the FinnGen study[16] was available for a subset of 273,765 individuals fulfilling similar inclusion criteria, of which 93% had received their first dose of a COVID-19 vaccination by 31 October 2021. The details of data preprocessing are reported in the Methods.

FinRegistry is a joint project of the Finnish Institute for Health and Welfare (THL) and the Data Science Genetic Epidemiology research group at the Institute for Molecular Medicine Finland, University of Helsinki. The FinRegistry project has received approvals for data access from the National Institute of Health and Welfare (THL/1776/6.02.00/2019 and subsequent amendments), Digital and Population Data Services Agency DVV (VRK/5722/2019-2), Finnish Center for Pension (ETK/SUTI 22003) and Statistics Finland (TK-53-1451-19). The FinRegistry project has received Institutional Review Board (IRB) approval from the National Institute of Health and Welfare (Kokous 7/2019).

### Income and medicine purchases correlate with COVID-19 vaccination uptake

We studied the importance of the 12 categories of predictors in predicting COVID-19 vaccination uptake using machine learning models (XGBoost[17]) trained separately for each category. Training was conducted in a randomly sampled 80% of the study population and evaluated in the remaining 20%. To speed up training, controls were downsampled in the training data so that five randomly sampled controls (vaccinated) were included for each case (unvaccinated). This downsampling did not significantly affect the XGBoost model predictions (Extended Data Fig. 2a). See the Methods and Extended Data Fig. 2b for more details on model training. Each model also included age and sex as predictors, representing the baseline model. Income (area under the receiver operating characteristic curve (AUC), 0.710; 95% confidence interval (CI), 0.708–0.712) and history of previous medication purchases, including 376 medications classes (AUC = 0.706; 95% CI, 0.704–0.708), were the most predictive categories (Fig. 2a and Supplementary Table 1). The CIs for AUC were computed using bootstrapping (Methods). All but one of the categories, long-term care, performed better than the simple baseline model including only age and sex (AUC = 0.612; 95% CI, 0.610–0.614; Fig. 2a, dotted line). Because many of the predictors highly correlate with age and sex, comparison with the performance of the baseline model shows how much additional predictive information the categories contain.

Next, we studied the classification performance of individual predictors within each category by training individual Lasso[18] models for each of the 2,890 predictors, including the baseline variables age and sex (Fig. 2b and Supplementary Table 2; see Methods for details). Lasso is a logistic regression model penalized with the L1 norm that acts as both a regularizer and a feature selector. The rationale was to establish a baseline that can be achieved using individual predictors. To provide interpretable effect sizes, we also performed logistic regression (without penalization) for each of the predictors, including age and sex as covariates, and calculated odds ratios (ORs) of not vaccinating against COVID-19 (Extended Data Fig. 3, Supplementary Table 3 and Methods). No downsampling of the training data was done for the individual predictor models. The Benjamini–Hochberg method was used to adjust the P values for multiple hypothesis testing. The reference levels for the predictors used in the logistic regression analysis are listed in Supplementary Table 8.

Not having income from labour in 2019 was the most predictive individual predictor (AUC = 0.668; 95% CI, 0.666–0.671; OR = 1.35; 95% CI, 1.35–1.35). Among individuals with labour income, those in the

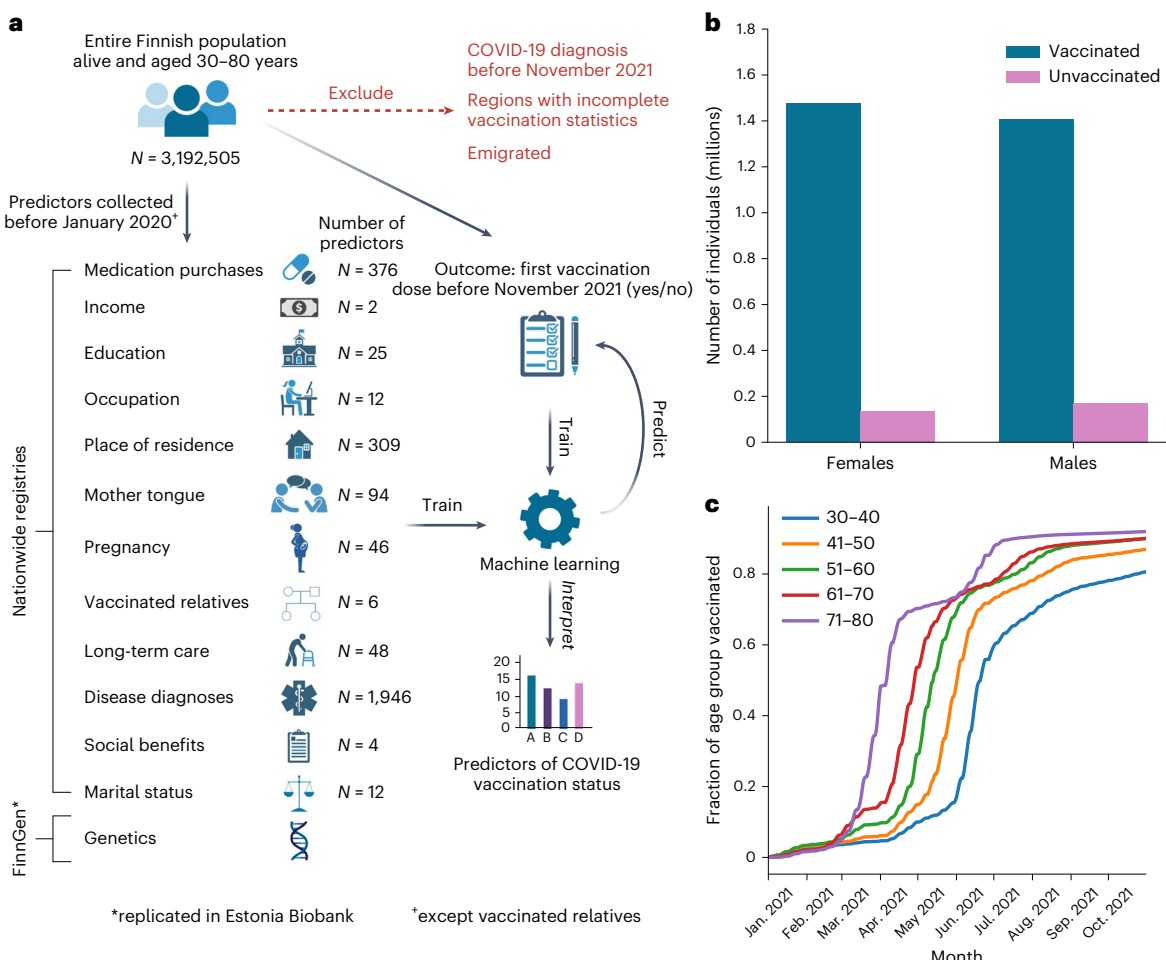

**Fig. 1 | Schematic outline of the study. a**, COVID-19 vaccination uptake (at least one dose) at the end of October 2021 was extracted from the Finnish Vaccination Register for each individual aged 30–80 years and living in Finland. A comprehensive collection of potential predictors was extracted (at the end of 2019, except for vaccination status of relatives, for which data up to the end of October 2021 were used) from nationwide registries, totalling 2,890 potential predictors across 12 manually defined predictor categories. The genetics of COVID-19 vaccination uptake was studied in a subsample of individuals of the total study population (FinnGen participants) and replicated in Estonia

Biobank. Machine learning was then used to identify the predictors and predictor categories that best predict vaccination uptake in the test set. **b**, Total number of vaccinated (blue, at least one vaccination dose) and unvaccinated (purple) females and males in the study population at the end of October 2021. **c**, Cumulative fraction of different age groups in the study population (blue indicates 30- to 40-year-olds, orange indicates 41- to 50-year-olds, green indicates 51- to 60-year-olds, red indicates 61- to 70-year-olds and violet indicates 71- to 80-year-olds) who had received the first dose of a COVID-19 vaccine as a function of time during the follow-up period. Panel **a** created with BioRender.com.

---

lowest income decile were less likely to take the vaccine than individuals in the 40–50% income decile bin (OR = 1.08; 95% CI, 1.08–1.09; Fig. 2c). Overall, we observed a linear relationship between income and COVID-19 vaccination uptake. Other socio-demographic variables such as speaking another mother tongue than Finnish or Swedish were strong predictors and conferred an elevated relative risk of not vaccinating (AUC = 0.649; 95% CI, 0.647–0.651; OR = 1.27; 95% CI, 1.27–1.27).

We examined individual disease diagnoses (Fig. 2d) and medication purchases to identify possible disease groups associated with vaccination uptake (Supplementary Table 3). The highest ORs of not vaccinating were observed for diagnoses of substance abuse, such as stimulants (OR = 1.22; 95% CI, 1.21–1.23) and cannabinoids (OR = 1.25; 95% CI, 1.24–1.26), and for hepatitis C diagnosis (OR = 1.22; 95% CI, 1.21–1.23), which is itself strongly associated with intravenous drug usage. Other mental health conditions, particularly those associated with psychotic-type or delusion-type symptoms, showed large relative risks (for example, OR of dissocial personality disorder, 1.24; 95% CI, 1.23–1.26; OR of schizoid personality disorder, 1.14; 95% CI, 1.13–1.15).

While medication purchase history was the second strongest predictor category, no single medication alone was a strong predictor, suggesting that the combined history of different medication purchases is largely responsible for the strength of the association. However, several of the most predictive medications associated with not vaccinating were those used in the treatment of psychosis-associated disorders, such as phenothiazines (OR = 1.07; 95% CI, 1.07–1.07) and novel/atypical antipsychotics (OR = 1.07; 95% CI, 1.07–1.07). Attention-deficit/hyperactivity disorder (ADHD) medications (centrally acting sympathomimetics) had the highest OR among the individual medications (1.08; 95% CI, 1.07–1.09). Memantine (other anti-dementia drugs), a medication used to treat symptoms of cognitive impairment such as Alzheimer's disease, was also associated with lower vaccination rate (OR = 1.04; 95% CI, 1.03–1.05).

Because we had comprehensive information on multi-generational familial relationships, we could study how the vaccination status of a close relative impacts the likelihood of vaccinating (Extended Data Fig. 4). We considered only individuals who had relatives in the study population (Supplementary Table 3 and Methods). We found that having an

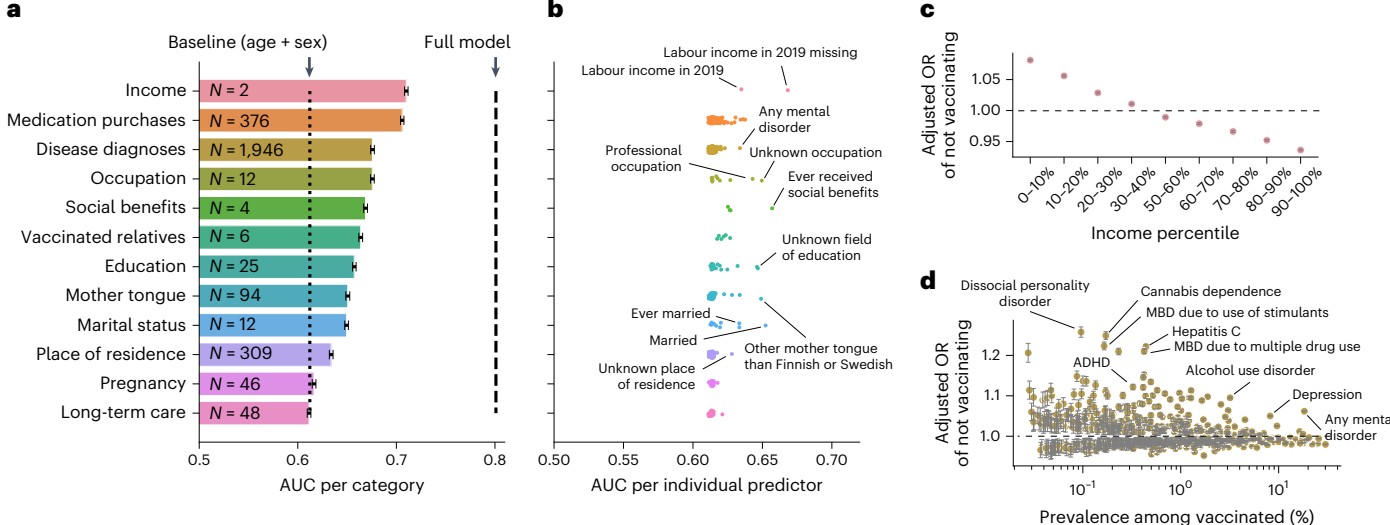

**Fig. 2 | Predictors of COVID-19 vaccination uptake. a**, AUC for XGBoost classifiers trained using predictors from different categories (each model also includes the baseline predictors age and sex). The error bars show 95% CIs computed using bootstrapping; the centres of the error bars correspond to the point estimates. The number of predictors in each category is indicated on the corresponding bar. The black dashed vertical line indicates the performance of the full XGBoost model using all predictors. The black dotted vertical line corresponds to an XGBoost model allowed to use only age and sex as predictors. Most of the predictor categories perform better than this baseline model, with income and medication purchases being the most predictive categories. **b**, AUC from Lasso classifiers trained separately for each of the individual predictors (the models also include the baseline predictors age and sex), grouped by the categories. Some of the highly predictive predictors have been highlighted (for a fully annotated list of AUCs of individual predictors, see Supplementary Table 2). **c**, Association between labour income in 2019 and COVID-19 vaccination uptake. The ORs are from a logistic regression model using income percentile bins as

predictors and adjusting for age and sex. The 40–50% percentile bin was used as a reference category. The dots represent point estimates of ORs. The error bars indicate 95% CIs for the ORs computed using bootstrapping. **d**, Associations between previous disease diagnoses and COVID-19 vaccination status. The ORs are from a logistic regression model using a binary disease indicator as the predictor and adjusting for age and sex. Some of the interesting predictors are highlighted. MBD, mental and behavioural disorders. Predictors with multiple hypothesis testing-adjusted $P > 0.01$ (Benjamini–Hochberg method) and prevalence among vaccinated <1,000 are not shown. $P$ values are two-sided and were calculated by dividing the coefficient values by their standard errors and observing the probability mass corresponding to equal or more extreme values from both tails of the standard normal distribution (as in the R package glm). The dots represent the point estimates of ORs. The error bars indicate 95% CIs for the ORs computed using bootstrapping. For a fully annotated list of the ORs of individual predictors, see Supplementary Table 3.

unvaccinated mother increases the odds of not vaccinating (OR = 1.31; 95% CI, 1.31–1.32) more than having an unvaccinated father (OR = 1.23; 95% CI, 1.22–1.23) or having any unvaccinated siblings (OR = 1.17; 95% CI, 1.16–1.17).

We performed a sensitivity analysis to account for possible non-reported emigration outside Finland. To capture unreported emigration, we excluded all individuals with no data entries in 2019 (4.0%; Methods). Overall, we did not observe differences in predictive performance for most individual predictors as measured by AUC (Extended Data Fig. 5a). However, we observed significant deflation in the ORs of several rare mother tongues (Extended Data Fig. 5b). The OR for speaking another mother tongue than Finnish or Swedish decreased from 1.27 to 1.15.

**A prediction model for COVID-19 vaccination uptake**
Combining all the registry-based predictors into a single XGBoost model provided good discrimination (AUC = 0.801; 95% CI, 0.799–0.803 in the test set; see also Fig. 2a) but modest calibration, with the predicted probabilities being higher than the observed non-vaccination rates. However, we recalibrated the model using the method from ref. 19 and obtained a better match between the predicted probabilities and the observed non-vaccination rates (Extended Data Fig. 6). In the test set, the top 1% of individuals with the lowest predicted probability to vaccinate (N = 6,385) had an observed vaccination rate of only 18.8% compared with 90.3% when considering everyone in the test set (Fig. 3a). The XGBoost classifier outperformed a Lasso classifier trained using the same full set of predictors (AUC = 0.778; 95% CI, 0.776–0.780).

We analysed the importance of each predictor in the combined XGBoost model by computing the mean absolute Shapley values of the predictors[20]. Income, the total number of medication purchases, age, the total duration of received social benefits and marital status were the most important predictors of COVID-19 vaccination status in this model (Fig. 3b). Interestingly, income was a more important predictor than age.

**Different predictor categories share similar information**
To study how much independent information each predictor category contains, we considered all possible combinations of predictor categories and trained a separate Lasso classifier model for each of the 4,097 combinations. By testing each possible combination of categories, we can quantify information relevant to COVID-19 vaccination prediction that is unique to single categories versus what is shared across categories.

Figure 4a shows the drop in AUC when each predictor category is removed separately from the combined model. As expected, classification performance decreased the most when we removed the medication purchases history category, leading to a drop in AUC of 1.3%. However, this decrease was substantially lower than the AUC improvement that this category contributed on top of age and sex (15.3%), indicating that much of the predictive information from this category was captured by other categories in the combined model.

We then studied the impact of removing multiple categories simultaneously on the prediction of COVID-19 vaccination uptake. This allowed us to identify category combinations that had the largest effect on the model predictions (Fig. 4b). For example, removing 10 of the 12

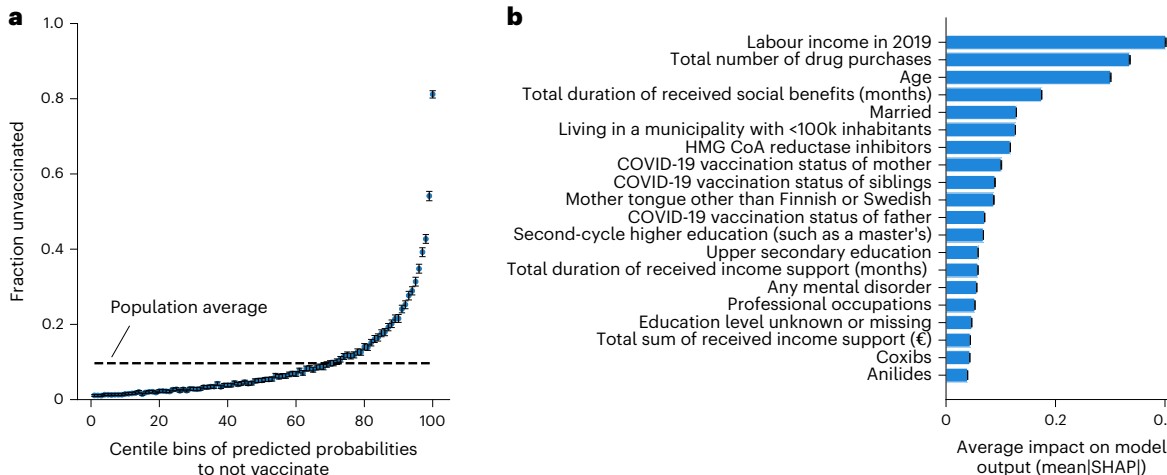

**Fig. 3 | A prediction model for COVID-19 vaccination uptake. a**, Fractions of unvaccinated individuals in the test set as a function of centile bins of predicted probabilities to not vaccinate from the full XGBoost model. The 99th centile bin comprises 6,385 individuals that have only an 18.8% (95% CI, 17.9–19.8%) chance of vaccinating. The error bars indicate 95% CIs computed using bootstrapping. The black dashed line indicates the average fraction of unvaccinated individuals in the study population. **b**, Mean absolute SHAP values[20] computed for all individual predictors used in the full XGBoost model. A higher value indicates a higher average impact of the predictor. The top 20 most important predictors are shown for clarity. The error bars indicate 95% CIs computed using bootstrapping (some uncertainty estimates are very small). HMG CoA, β-hydroxy β-methylglutaryl coenzyme; A, Coxibs, cox-2 inhibitors.

categories resulted in only a 7.4% AUC decrease (with the most predictive model containing two categories, occupation and medication purchases) compared with the full model. Taken together, these results indicate that predictive information was shared across categories and relatively good prediction accuracy can be achieved even in settings where some of the information used in this study is missing.

To understand how much predictive information was shared across categories, independently of age and sex, we computed partial pairwise Pearson correlation between predicted probabilities obtained from models trained separately in each category (Fig. 4c; the same models are shown in Fig. 2a). We found that COVID-19 vaccination uptake probabilities predicted using the income, education, occupation and social benefits categories were highly correlated and clustered together (Pearson partial correlation coefficient, >0.25). We also identified significant correlations between predicted probabilities from socio-economic categories and health-related categories. For example, the correlation between predicted probabilities from the income and medication purchase history categories was 0.22.

### Genetic information is a weak predictor of COVID-19 vaccination

We performed a genome-wide association study (GWAS) of COVID-19 vaccination uptake in FinnGen ($N$ = 273,615; ~8.6% of FinRegistry participants aged 30–80) and the Estonian Biobank ($N$ = 145,615), restricted to recent European ancestry. The effects were consistent across the two studies as evidenced by a genetic correlation of 0.8 (95% CI, 0.66–0.95). We therefore performed a meta-analysis using METAL[21]. We identified eight genome-wide significant loci ($P \leq 5 \times 10^{-8}$) (Fig. 5a and Methods), and, in Supplementary Table 5, we report the most likely gene linked to each lead variant by using a machine-learning-based prioritization score from Open Targets Genetics[22,23]. Four of the eight lead variants were associated with anthropometric traits, such as body fat distribution (Supplementary Table 5). These four variants increased the likelihood of vaccination while being associated with reduced body fat. We next investigated the single-nucleotide-polymorphism-based (SNP-based) heritability (the fraction of phenotypic variance in the population explained by the additive effects of SNPs, not to be confused with genetic influence) of vaccination uptake through linkage disequilibrium score regression[24], finding a low but statistically significant SNP-based heritability (observed scale $h^2_{SNP}$ = 2.6%, s.e. = 0.18%, $P = 1.36 \times 10^{-47}$).

Given the significant heritability, we explored whether we could build a polygenic score (PGS) for COVID-19 vaccination uptake. We reran the GWAS on 70% of the FinnGen individuals, meta-analysed these results with the GWAS conducted in the Estonia Biobank and used the results to build a PGS in the remaining 30% of the FinnGen individuals. A model including age, sex and the PGS reached an AUC of 0.612 (95% CI, 0.601–0.623) when predicting vaccination uptake, significantly higher than the baseline model including only age and sex (AUC = 0.589; 95% CI, 0.578–0.600; $P$ for improvement, $1.72 \times 10^{-9}$). The PGS predicted vaccination status better than the pregnancy and long-term care categories, and at a similar accuracy to municipality of residence (Supplementary Table 1).

We explored the genetic correlations between the GWAS of vaccination uptake and a series of other health and behavioural information, mostly not available in the nationwide FinRegistry dataset. Of the 23 phenotypes tested, 11 were significant after multiple hypothesis testing correction ($P < 2 \times 10^{-3}$, Bonferroni corrected for 23 tests; Fig. 5b). Four psychiatric disorders—schizophrenia, major depressive disorder, bipolar disorder and ADHD—were positively genetically correlated with reduced vaccination uptake ($r_g$ between 0.18 and 0.43), consistent with the epidemiological results (Fig. 2c). Not vaccinating was also associated with a higher genetic predisposition to loneliness, risky behaviour and smoking ($r_g$ between 0.25 and 0.33). Interestingly, we found a negative correlation ($r_g$ = −0.34; 95% CI, −0.40 to −0.28) with participation in subsequent questionnaires of UK Biobank (a proxy for engagement in scientific research) (Supplementary Table 6). Genetic correlations were comparable when COVID-19 cases were included in the vaccination uptake phenotype (Extended Data Fig. 7; only for the FinnGen study).

To test whether individuals at higher genetic risk for COVID-19 critical illness, hospitalization and susceptibility were more or less likely to vaccinate, we built a PGS for each of the three COVID-19 phenotypes using Release 7 from the COVID-19 Host Genetics Initiative[15], which includes mostly studies collected before the start of the vaccination campaigns. Individuals with higher PGSs for each COVID-19 phenotype were less likely to receive the vaccine. However, the association was modest (critical illness: OR = 1.02; 95% CI, 1.01–1.04; hospitalization:

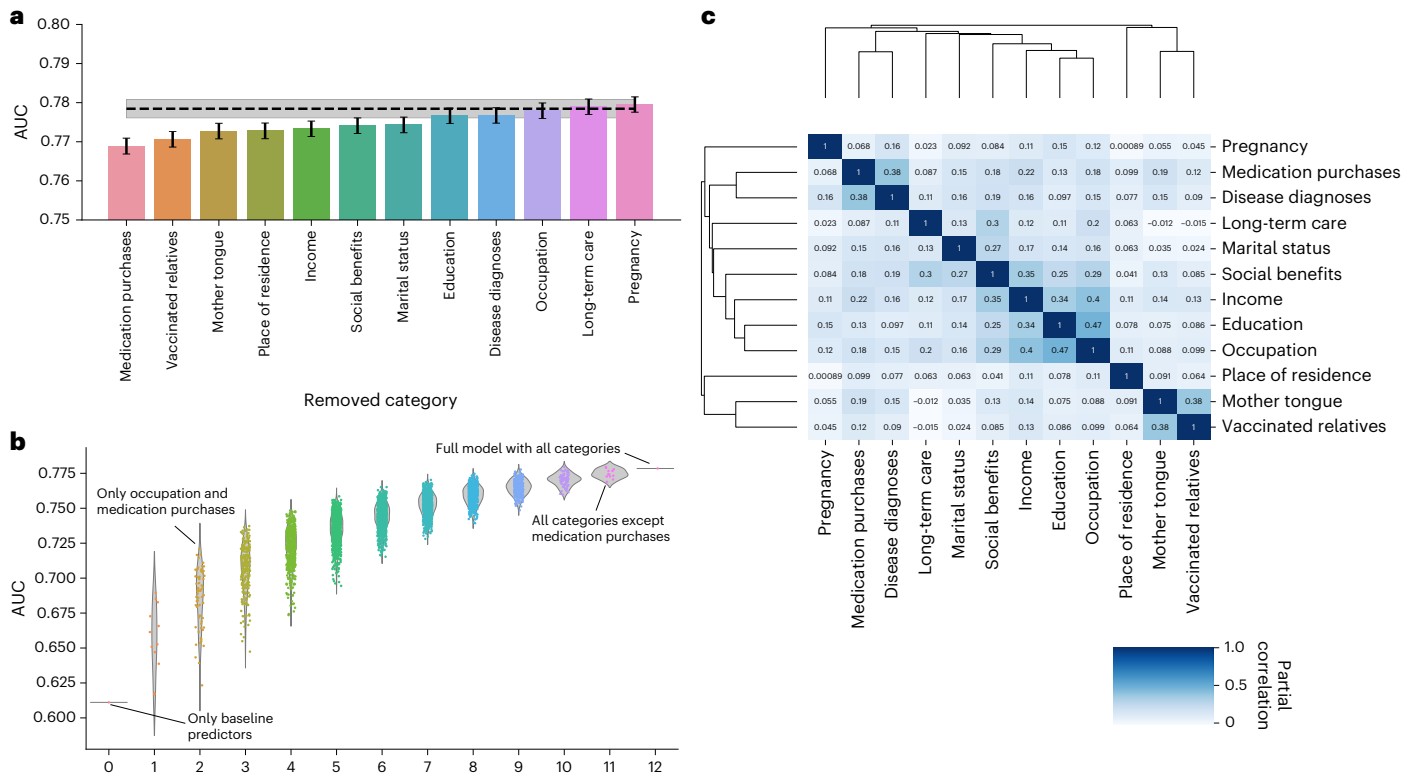

**Fig. 4 | Shared information across different predictor categories. a**, Drop in AUC (*y* axis) when removing a single category at a time from the full Lasso classifier (including all predictors). Removing all predictors from a category removes all information unique to the predictors of that category, meaning that the drop in AUC quantifies the loss in predictive power due to information unique to the removed category. The lower the AUC here, the higher is the amount of unique information contained in the category that is useful for predicting COVID-19 vaccination uptake. The black dashed line indicates the AUC of the full Lasso model using all predictor categories. The error bars and the error band correspond to 95% CIs computed using bootstrapping; the centres of the error bars correspond to the point estimates. **b**, Drop in AUC (*y* axis) when removing different combinations of predictor categories from the full Lasso model (the full model corresponds to 'Number of included categories = 12'). All combinations of removed categories were tested by training separate Lasso classifiers on the data including only the specific combination of predictor

categories, and the corresponding AUCs are shown as individual dots. The violin plots show the distribution of AUCs for each number of removed categories. Individual models discussed in the text are highlighted and named. The model with zero removed categories corresponds to a model trained using the baseline predictors age and sex only. All models include age and sex as predictors. Panel **a** shows a detailed view of 'Number of included categories = 11'. **c**, Pairwise partial Pearson correlation, adjusting for age and sex, between predicted probabilities of COVID-19 vaccination uptake for each test set sample, obtained from each category separately (XGBoost classifiers; the AUCs for these models are shown in Fig. 2a and Supplementary Table 1). The colour indicates the strength of correlation, and the correlation coefficient is shown on each heat-map cell. Hierarchical clustering dendrograms of the partial correlation matrix of model predictions are shown beside the matrix and were used in ordering the rows and columns.

OR = 1.04; 95% CI, 1.02–1.05; susceptibility: OR = 1.02; 95% CI, 1.01–1.04 per standard deviation in PGS), partially due to the PGS for COVID-19 being weakly associated with COVID-19.

Using Mendelian randomization (MR), we found no evidence of a causal relationship between COVID-19 phenotypes and vaccination uptake (critical illness: MR inverse-variance weighted effect (IVW), 0.015; s.e. = 0.019; *P* = 0.44; hospitalization: MR IVW, 0.018; s.e. = 0.028; *P* = 0.52; susceptibility: MR IVW, −0.017; s.e. = 0.069; *P* = 0.81) (Supplementary Table 9). Similarly, we did not observe a causal relationship of height or type 2 diabetes with vaccination uptake. However, higher body mass index (BMI) was causally related to decreased vaccination uptake (MR IVW, 0.121; s.e. = 0.033; *P* = 2.1 × 10$^{-4}$), with no evidence of unbalanced pleiotropy (MR Egger intercept, −1.3 × 10$^{-3}$; s.e. = 1.5 × 10$^{-3}$; *P* = 0.39).

## Discussion

The digitalization, harmonization and accessibility of information collected by health care organizations and by governmental agencies can inform policymakers at an unprecedented breadth. The comprehensive

collection of nationwide registers combined with biobank data and empowered by machine learning approaches allowed us to extensively compare the correlations of health-related, socio-economic, familial, genetics and demographic information with one of the most pressing public health issues: participation in COVID-19 vaccination programmes.

Even in the relatively economically equal Finnish society (top 15 in income equality among all countries[25]), socio-economic aspects and labour income in 2019 were the strongest predictors of receiving the first dose of a COVID-19 vaccine. This observation could be partly explained by people in lower-income occupations having limited access to vaccines due to their stricter working schedules. Nonetheless, information about professions was a weaker predictor of vaccination uptake than income. The lack of income in 2019, the strongest predictor, captures a wide range of socio-economic factors including unemployment, severe illness and retirement.

Several disease-related conditions were associated with vaccination uptake. Mental health issues were the most important category: psychosis-related conditions and diagnoses related to substance use

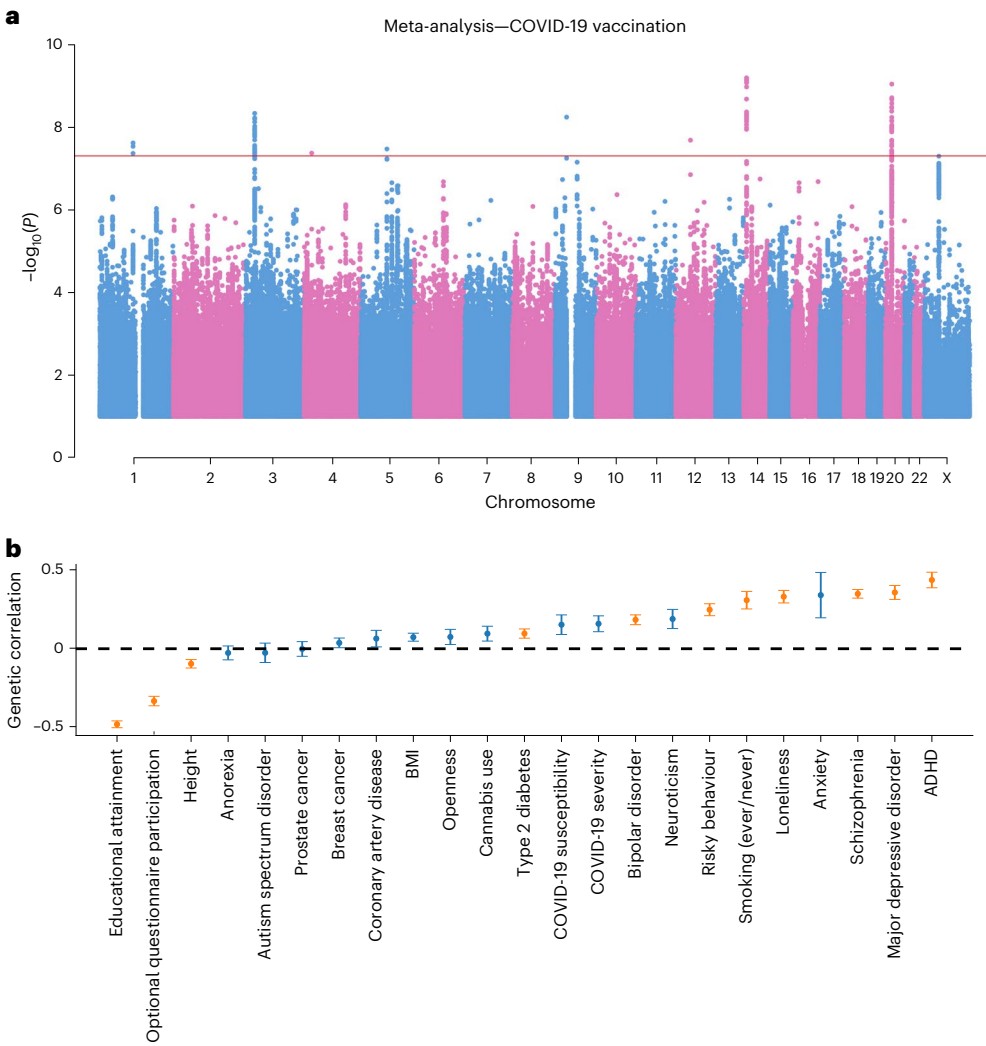

**Fig. 5 | Genetic correlates of COVID-19 vaccination uptake. a**, Manhattan plot of COVID-19 vaccination uptake from a meta-analysis of FinnGen and the Estonian Biobank. Genetic variants must have been tested in both datasets and passed quality control in both (INFO ≥ 0.8 and MAF ≥ 0.1%), and significant variants must not have indicated significant heterogeneity (heterogeneity $P < 0.0056$; the $P$ values were Bonferroni corrected for multiple testing with nine significant variants). The red horizontal line indicates genome-wide significance. **b**, Genetic correlations between COVID-19 vaccination uptake and selected health and behavioural phenotypes. The point estimates represent correlations, and the error bars reflect standard errors. Orange error bars and point estimates represent Bonferroni-significant genetic correlations ($P < 0.002$, Bonferroni corrected for multiple testing with 23 tests). The black dashed line indicates zero genetic correlation. For both panels, the $P$ values are two-sided and were calculated by dividing the coefficient or correlation values by their standard errors and observing the probability mass corresponding to equal or more extreme values from both tails of the standard normal distribution (as in the R package glm). A positive correlation means a correlation with reduced COVID-19 vaccination uptake.

disorders were associated with lower vaccination uptake. Associations with individual medication purchases supported these observations. People with mental health disorders are at increased risk of (severe) COVID-19, but even more notably, COVID-19 can cause deterioration in mental health, reduction in neuropsychiatric functioning and even neurodegeneration[26]. As those suffering from mental health disorders appear to be less likely to receive a vaccine, efforts to increase their vaccination uptake could prove especially effective in reducing both the acute infections and the multifactorial burden of long COVID.

Interestingly, medications used in the management of Alzheimer's and Parkinson's diseases were associated with lower vaccination rates. People with these conditions are at higher risk of severe COVID-19 (ref. 27), probably have reduced functioning in everyday life and are less able to make informed decisions about their vaccination. Other, more common diseases were associated with reduced vaccination uptake, probably by capturing underlying socio-economic factors.

Previous studies have shown that the experience of a family member with COVID-19 increased acceptance of the COVID-19 vaccine[28]. In line with this observation, we found that vaccination status correlates within families. For example, having an unvaccinated mother correlated with the rate of not being vaccinated (OR = 1.31; 95% CI, 1.31–1.32). However, having an unvaccinated mother has a stronger association with reduced vaccination than having an unvaccinated father or unvaccinated siblings, indicating that other factors beyond those shared within families (for example, socio-economic status) influence the correlation with vaccination status.

History of medication purchases was the strongest predictor category alongside income, despite none of the individual medicines being a strong predictor alone. We hypothesize that the pattern of medication purchases is a relatively good proxy for both health and socio-economic aspects. Considering this observation, we performed extensive analyses to understand whether different predictor categories are capturing

overlapping information. We found a large overlap and redundancy in the predictive properties of different categories, some of which are traditionally considered independently (for example, health and socio-economic indicators). This observation is important for two reasons. First, it blurs the distinction between health and socio-economic information. The overlap between these two categories has implications for law and ethics. For example, informed consents in biomedical studies are often bound to health-related research, while we show that socio-economic and health information can capture similar underlying aspects in predicting vaccination uptake. Second, it questions the feasibility of excluding information perceived as sensitive from machine-learning-based prediction models. For example, citizens might be against using income to identify individuals at higher risk of not vaccinating but be more inclined to accept targeting individuals on the basis of certain previous health conditions. We have shown that predicted probabilities of vaccination uptake obtained using medication purchase history are correlated with predictions obtained using income, questioning whether medication purchase information should be used in a hypothetical scenario where income cannot be used as a predictor. Future work using dimensionality reduction techniques to obtain latent factors might help reduce information redundancy and identify some of underlying attitudinal aspects that are not directly measured in the registers.

We showed that by including all 2,997 predictors we could train a model of COVID-19 vaccination uptake. For example, such a model can be used to identify 1% of the population with an average vaccination rate of approximately 19%, which is almost five times lower than the national average.

GWASs have been conducted on thousands of health and behavioural traits, and many behavioural traits correlate with genetic variants[29]. Importantly, both COVID-19 susceptibility and severity correlate with genetic variants[15]. This, together with the observation that many of the registry-based predictors in this study have genetic correlates (for example, mental health disorders), prompted us to study genetic associations with COVID-19 vaccination uptake.

Genetic information has been measured in nearly 10% of the Finnish population as of 2023. It has low measurement error, is stable through life and is not impacted by reverse causation. For these reasons, statistical genetics approaches can be used to identify correlates of vaccination uptake that cannot be easily measured nationwide. Such correlates can be tested for replication in datasets from other countries. We demonstrate this by performing a meta-analysis on FinnGen and Estonian Biobank study participants and showing that genetic correlations between vaccination uptake and socio-economic traits or psychiatric disorders persist across countries. Interestingly, we found a significant genetic correlation with participating in optional questionnaires within the UK Biobank, supporting a shared underlying effect between participating in scientific studies and propensity to vaccination. As is the case for many complex traits, the significant SNPs identified were robustly associated with vaccination uptake, but the effect sizes were very small in isolation. Summing these effects into a PGS produced a weak predictor of COVID-19 vaccination. Finally, our results indicate that individuals at higher genetic risk of severe COVID-19 were less likely to be vaccinated but that this association was not causal and was more likely due to shared risk factors captured by the PGS.

Our approach has several limitations. First, generalizability outside Finland and to non-European ancestries is unclear, and replication in other countries is needed to understand the generalizability of our findings across different populations. Generalizability can be impacted by differences in disease prevalence (for example, schizophrenia being more common in Finland than in most European countries[30]) but also because of varying methods of organizing national vaccination programmes across vulnerable groups. Similarly, genetic associations have been shown to lack portability across ancestries due

to differences in minor allele frequency (MAF) and linkage disequilibrium[31]. Studies including individuals from additional ancestries are needed before generalizations on the genetic results can be made. Previous studies using nationwide registers have, however, shown similar risk factors for severe COVID-19 as in other countries[32,33]. Second, information about deaths and emigration from Finland during the year 2021 was not available to us. Thus, some individuals might not have taken the COVID-19 vaccination because they had passed away or had emigrated during the follow-up period. We restricted the analyses to individuals younger than 80 years old to reduce the number of individuals expected to die in 2021. Third, due to the scope and complexity of the included predictors, we made some simplifying decisions in preprocessing the nationwide registry data. The predictors included in the analyses are thus subject to some simplifications and limitations. We considered disease diagnoses and medication purchases over the lifespan of the individuals in the study population and condensed this information into binary yes/no predictors. Missing values for many socio-economic variables were considered by including separate predictors for missingness, but there might be multiple reasons for missing records. Better modelling of missing data and age of diagnosis would probably further increase the predictive performance of the models presented in this study. Not everyone reports emigrating outside Finland to the authorities. To capture this potential bias, we performed a sensitivity analysis removing individuals with no data entries in the year 2019 and showed no significant changes overall in the AUCs of individual predictors.

In conclusion, by performing a nationwide examination of predictors of COVID-19 vaccination uptake across different life domains, we highlight the importance of harmonized and accessible registry and biobank-based information. We have shown that COVID-19 vaccination uptake is multifactorial and that individuals at higher risk of suffering the worst consequences of COVID-19 are also those with the lowest rates of COVID-19 vaccination. Using a machine learning approach, we could identify undervaccinated groups of individuals relative to the average population. A recent study has shown that financial incentives for vaccination do not have negative unintended consequences[34]. Targeting financial incentives on the basis of vaccination probabilities could be a cost-effective measure to increase the effectiveness of immunization programmes. While we cannot know whether targeting individual predictors is likely to change vaccination uptake, as our study does not address causality, the predicted probabilities from the model could be used to identify individuals at higher risk of not vaccinating. This could help better target existing awareness campaigns or financial incentives.

## Methods

We assert that all procedures contributing to this work comply with the ethical standards of the relevant national and institutional committees on human experimentation and with the Helsinki Declaration of 1975, and as revised in 2008. The FinRegistry project has received approvals for data access from the National Institute of Health and Welfare, DVV (Digi- ja väestötietovirasto), the Finnish Center for Pension and Statistics Finland. The FinRegistry project has received IRB approval from the National Institute of Health and Welfare.

Patients and control participants in FinnGen provided informed consent for biobank research, on the basis of the Finnish Biobank Act. Alternatively, separate research cohorts, collected before the Finnish Biobank Act came into effect (in September 2013) and the start of FinnGen (August 2017), were collected on the basis of study-specific consents and later transferred to the Finnish biobanks after approval by Fimea (Finnish Medicines Agency), the National Supervisory Authority for Welfare and Health. Recruitment protocols followed the biobank protocols approved by Fimea. The FinnGen study is approved by the Finnish Institute for Health and Welfare, the Digital and Population Data Service Agency, the Social Insurance Institution, Findata, Statistics Finland and the Finnish Registry for Kidney Diseases.

The activities of the Estonian Biobank are regulated by the Human Genes Research Act, which was adopted in 2000 specifically for the operations of the Estonian Biobank. Analysis of individual-level data from the Estonian Biobank was carried out under ethical approval no. 1.1-12/3022 from the Estonian Committee on Bioethics and Human Research (Estonian Ministry of Social Affairs), and according to data release application 3-10/GI/31487 from the Estonian Biobank, Institute of Genomics, University of Tartu.

### Study population

The FinRegistry dataset (https://www.finregistry.fi/), used in the phenotypic analyses, includes 7,166,416 individuals of whom 5,339,804 (74.51%) are index individuals (every resident of Finland alive on 1 January 2010). The remaining 1,826,612 individuals are relatives (offspring, parents and siblings) and spouses of index individuals who are not index individuals themselves.

FinnGen[16] and the Estonian Biobank[35] were used to explore the role of genetics in COVID-19 vaccination status. The FinnGen project combines multiple hospital biobanks and digital health registries. The data release used for this analysis (Release 9) has genotype data available for 377,277 individuals of Finnish ancestry. The Estonian Biobank is a volunteer-based sample with continually updated national health registry linkage and genotype data on 202,910 individuals.

To restrict the study population to individuals who had had a fair opportunity of receiving the first dose of a COVID-19 vaccination by the end of October 2021, we excluded the following individuals:

1. Individuals who had died or emigrated before 31 December 2020 (death statistics for 2021 in Finland were not available).
2. Individuals who were less than 30 years old on 31 October 2021.
3. Individuals who were more than 80 years old on 31 October 2021.
4. Individuals who had a laboratory-confirmed COVID-19 diagnosis prior to 31 October 2021.
5. Individuals living in the municipality of Askola.

For the genetic analyses conducted in FinnGen and the Estonian Biobank, death or emigration was limited to 31 December 2019, as statistics beyond this date were unavailable for FinnGen. Residents of Askola were excluded because it was the only municipality where the vaccination coverage differed radically from any other Finnish municipality (Extended Data Fig. 1). After these exclusion criteria, the final FinRegistry study population contained 3,192,505 individuals, and the FinnGen study population included 273,615 individuals.

The study outcome, having received at least one dose of a COVID-19 vaccine by 31 October 2021, was defined for the Finnish data using the official registry-based definition by the National Institute for Health and Welfare:

1. Identifying all participants with a record for the anatomical therapeutic chemical (ATC) code J07BX03 (COVID-19 vaccinations).
2. Identifying all participants with a record corresponding to a relevant drug name. The criteria included all records with a drug definition or trade name including 'COM', 'COV', 'CVID', 'CO19', 'COR', 'KOR', 'PFI', 'MOD', 'AST', 'AZ', 'BION' or 'SPIKE'. From the set of records identified using these criteria, we excluded ambiguous records containing 'TIC', 'ZOSTA', 'NEULA', 'VESIROK', 'DUKORAL', 'TUHKA', 'COVAC', 'VAZ', 'ZAST', 'PASTEUR', 'FLUR', 'LASTEN', 'KURKKU' or 'SUSTA'.
3. Records were considered only after 1 October 2020.

In the Estonian Biobank, the study outcome of having received at least one dose of a COVID-19 vaccine was defined on the basis of linked data from the national Health and Welfare Information Systems Centre (TEHIK). Health care providers in Estonia have to submit all vaccination notifications to TEHIK, which is also the institution responsible for creating vaccination certificates. The database contains the following information: name of vaccine, ATC code, amount (mcg), dosing and schedule. We included all individuals with at least one record of a COVID-19 vaccine (ATC code J07BX03) between 10 October 2020 and 31 October 2021 as cases and others as controls. The analyses were not performed blind to the vaccination status.

### Selection and definition of the phenotypic predictors

The FinRegistry study contains a comprehensive selection of data modalities ranging from disease history to medication purchase history and detailed socio-economic variables, as illustrated in Fig. 1a. We performed an initial variable selection by manually curating variables of interest across the different registries. Categorical variables were dichotomized into indicator variables. Individual predictors and their manually curated categories are listed in Supplementary Tables 2 and 3. Division into the 12 predictor categories was based on expert knowledge and largely reflects the source registers of the predictors—the FinRegistry dataset is aggregated from several individual thematic registers. For example, the social benefits category predictors come from the Finnish Register of Social Assistance, which is a collection of variables detailing received periods of income support.

For each predictor, excluding disease occurrences and medication purchases, we also included a binary predictor indicating whether the value for this predictor was missing. For disease diagnoses and medication purchases, not having a record of the diagnosis or purchase was interpreted as absence of the diagnosis or purchase. Taken together, we defined 2,997 predictors (including age and sex). The prevalence of the predictors within the study population was not assessed beforehand. To preserve the privacy of individuals in the study population, FinRegistry has a policy that allows exporting aggregated data only when the aggregated data are based on five or more individuals. Some of the very rare predictors had fewer than this number of individuals among either vaccinated or unvaccinated, and thus predictor-level results for these predictors could not be exported from the secure analysis environment. In total, 105 of the defined predictors were excluded from the predictor-level results due to this, leaving us with 2,892 predictors (including age and sex). The preprocessing of each category of phenotypic predictors is discussed in more detail below.

**Medication purchases.** Information about medication purchases was retrieved from the Social Insurance Institution of Finland, Kela, which is a government agency that provides basic economic security through financial support for Finnish residents and many Finns living abroad. One of the social security benefits provided by Kela is reimbursements of part of the costs of medicines that are prescribed for the treatment of an illness. These data contain nationwide information about prescribed medications that are purchased from pharmacies. They do not include medications delivered in hospitals or purchases of medications without a prescription. This register exists from 1995. We coded medication purchase information into binary predictors describing whether an individual ever purchased the medication during 1995–2019. Similar medications were collapsed into one predictor by considering only the first five digits of the ATC codes.

**Occupation.** Information about job occupation was retrieved from Statistics Finland, which is a Finnish public authority that collects, combines and stores data on a wide range of topics. Occupation is available for employed people at the end of the statistical reference year. The information exists from years 1970, 1975, 1980, 1990, 1993, 1995, 2000 and 2004 on an annual basis. We defined occupation as the latest reported (not unknown) occupation before 31 December 2019. Occupation information was coded into 11 binary predictors, according to the highest-level categorization in the Statistics Finland data.

**Disease history.** Disease history was captured using two sets of data: FinnGen clinical endpoints and the Finnish National Infectious Diseases Register. The clinical endpoints were originally defined for the FinnGen project[16] by a group of clinical experts. The clinical endpoints were predominantly generated by combining ICD8, ICD9 and ICD10 codes retrieved from the Finnish Institute of Health and Welfare registries (hospital discharge, cause of death and cancer registers). In addition, for a small proportion of clinical endpoints, information about medication purchases (Kela), medication reimbursements (Kela), surgical procedures (Finnish Institute of Health and Welfare) and primary health care ICD codes (Finnish Institute of Health and Welfare) was used. Clinical endpoints were filtered by excluding endpoints with fewer than 1,000 individuals in the FinRegistry population and redundant and highly correlated clinical endpoints as defined by FinnGen. Clinical endpoints defined solely on the basis of ATC codes were also excluded, as they capture the same information as medication purchases. For more information about FinnGen clinical endpoints and their definitions, see https://www.finngen.fi/sites/default/files/inline-files/FinnGen_Endpoints_Elisa%20Lahtela.pdf and https://risteys.finregistry.fi/. Clinical endpoints were collected between 1 January 1969 and 31 December 2019.

The Finnish National Infectious Diseases Register, retrieved from Finnish Institute of Health and Welfare, is based on the Communicable Diseases Act and Decree, which requires medical doctors and laboratories to report cases of certain infectious diseases. The data exist from 1995 to 2021. The ten most frequently reported infectious diseases were included as binary variables (having ever had the diagnosis before 31 December 2019), excluding COVID-19. COVID-19 diagnoses (up until the end of the study period, 31 October 2021) from the infectious diseases register were used to exclude people from the study population, as people with a COVID-19 diagnosis had different eligibility criteria for vaccination from the rest of the population. In total, the diseases category included 1,959 binary predictors that describe whether the individual has ever had the diagnosis.

**Income.** Information about income was retrieved from the Finnish pension registry. Income covers salary from labour, not income from benefits or capital income. Income from 2019 was used as a continuous predictor. The year 2019 was selected because it was the latest full year before the outbreak of the COVID-19 pandemic in Finland. Individuals with missing income information from 2019 ($N = 1,173,047$) were treated as missing data and were not included in computing the income percentiles in Fig. 2c. Missing income information was treated as a separate binary predictor. There are multiple reasons why income information might be missing, including unemployment, severe illness and retirement.

**Education.** Information about education level and field of education was retrieved from Statistics Finland as the highest completed degree by statistical year. The data exist for 1970, 1975, 1985 and every year between 1987 and 2018. Education level was defined as the highest completed degree by the end of 2018, and the field of education used was the field corresponding to the highest completed degree. Education level was coded into ten binary predictors, according to the highest-level categorization in the Statistics Finland data, except for adding one predictor corresponding to possibly ongoing education. Each individual aged between 30 and 35 was assigned to this category on the basis of the median age of receiving a doctoral degree in our dataset. Correspondingly, the field of education was set to 'education possibly ongoing' for everyone aged between 30 and 35. In total, the field of education was coded into 13 binary predictors.

**Marital status.** Information on marital status in the study population was retrieved from the Finnish Population Registry from the Digital and Population Data Services Agency. The data exist between 1960 and 2019. Marital status was coded into nine binary predictors using the latest known marital status. In addition, separate predictors for ever having been married or ever having been divorced were defined on the basis of the same original data.

**Social benefits.** The amount and duration of social benefits received were retrieved from the Finnish Register of Social Assistance. This register covers the years between 1985 and 2019 and includes social benefits received by social service clients who, due to the lack or insufficiency of income or social security benefits, have claimed social assistance. Social security benefits are not included in the social benefits category in this study.

The social benefits data used in this study are a combination of recipients of primary social assistance, preventive social assistance and rehabilitative work benefits. The social benefits category includes four predictors: total actual income support in euros received by an individual between 1985 and 2019, total number of months an individual has received actual income support in that same interval, total number of months an individual has received any income support and whether an individual has ever received social assistance.

**Long-term care.** The Care Register for Social Welfare from the Finnish Institute of Health and Welfare was used to obtain information about long-term care periods. This register contains data on activities and clients of institutional care and residential services of social welfare, and it covers the years between 1995 and 2019. The register contains comprehensive data from individuals who have been clients in private or public retirement homes, elderly 24-hour residential accommodation, institutional care and assisted living for the intellectually disabled, 24-hour residential housing for the severely physically or intellectually disabled, treatment for substance abuse, and rehabilitation facilities or non-round-the-clock housing services.

In this study, we used these data to create two sets of binary predictors. The first set contains 20 predictors that detail the type of care given to an individual (for example, living in an elderly home or rehabilitation facility). The second set contains 29 predictors that describe the main reason for entering the treatment. In addition, we created a predictor describing whether an individual had any periods of long-term care between 1995 and 2019 and another predictor to sum up the total number of treatment days within the same period.

**Place of residence.** The latest known place of residence was extracted from the Finnish Population Registry (Digital and Population Data Services Agency) on a municipality level. All individuals living in the municipality of Askola were discarded because the vaccination coverage in Askola was a heavy outlier. Place of residence was thus encoded as 306 binary predictors, including a predictor describing whether the place of residence was unknown.

**Mother tongue.** Information about mother tongue was obtained from the Finnish Population Registry from the Digital and Population Data Services Agency. This information is available between 1960 and 2019. Each mother tongue was considered as a separate binary predictor. Additionally, a predictor summarizing all mother tongues other than Finnish and Swedish was created.

**Pregnancy.** Information about pregnancy-related variables was obtained from the Medical Birth Register from the Finnish Institute of Health and Welfare. The information is available for all births in Finland between 1987 and 2019. We manually selected a set of 47 predictors from the Medical Birth Register. It is worth noting that the pregnancy-related information was used only for women who have been pregnant.

**Vaccinated relatives.** Information about COVID-19 vaccination status was obtained by combining the vaccination registry with information

about familial relationships within the study population retrieved from the Finnish Population Registry from the Digital and Population Data Services Agency. The familial information is available between 1964 and 2019.

For each individual in the study population, we created separate binary predictors describing the vaccination status of their mother and father. If the mother or father was not included in the study population, the value of the corresponding predictor was marked as missing. There are several reasons why a person's relative would not be included in the study population. They could be too young (<30 years), too old (>80 years), dead or emigrated.

We also created a binary predictor describing the vaccination status of possible siblings of each individual in the study population. The value of this predictor was coded as 0 if the individual had siblings and any of them was vaccinated, as 1 if the individual had siblings and none of them was vaccinated, and as missing if the individual had no siblings or if information about possible siblings' vaccination status was not available.

### Train/test split and imputation of missing values

The study population was divided at random into training and test sets. The training set contained 80% of the study population. Only the training set was used in model training and fitting. The test set was reserved for computing the performance of the models. Classification performance was measured using AUC, and the uncertainty of the obtained AUC values was estimated using bootstrapping by drawing 2,000 samples from the test set (with replacement) and computing the 95% CIs. To speed up the training of the Lasso and XGBoost classifiers, the training set was downsampled to include all of the non-vaccinated individuals (308,594) and four randomly sampled vaccinated individuals per non-vaccinated individual.

Each binary predictor category (except for medication purchases and disease diagnoses, as described above) includes a binary predictor that encodes whether the value was missing in the registries. For example, education level is encoded with nine binary predictors describing the education levels and one binary predictor indicating whether information about education level was missing. In the logistic regression analysis, individuals with missing values were discarded from the analysis. This corresponds to a complete case analysis. The number of missing values is shown for each predictor in Supplementary Tables 2 and 3. In the Lasso analyses, imputation was used to keep the dataset sizes constant across the compared predictors. Imputation was conducted by drawing new values for the missing values with replacement from the distribution of the non-missing values of the same predictor, assuming that the values are missing at random. In the XGBoost analyses, missing values were input to the algorithm as is, letting XGBoost learn the rules for handling missing values.

### Logistic regression

Logistic regression adjusted for age and sex was used to determine the association of each binary predictor with vaccination status (1, not vaccinated; 0, vaccinated). For each binary predictor, the following model was fit on the training split of the data using the function bigglm from the library biglm (v.0.9.2.1)[36,37]:

$$\text{vaccination status} \sim \text{age} + \text{sex} + \text{predictor}.$$

The reference categories for the different predictors are detailed in the Supplementary Table 8. The P values of the logistic regression model coefficients were corrected for multiple hypothesis testing using the Benjamini–Hochberg procedure[38], implemented in the Python package statsmodels (v.0.12.2)[39].

### XGBoost classifiers

XGBoost (eXtreme Gradient Boosting, v.1.5.0)[17] classifiers were trained for each predictor category and for the full set of predictors

to understand how much learning interactions and nonlinearities can boost the vaccination status predictions. All models were trained on the training split of the data using fivefold cross-validation to optimize the model hyperparameters using Bayesian hyperparameter optimization (BayesSearchCV function from scikit-optimize, v.0.9.0) over the range of possible hyperparameter values detailed in Supplementary Table 4, sampling 200 hyperparameter combinations for each model. Balanced class weighting was used to penalize the misclassification of both vaccinated and unvaccinated equally. Classification performance was marginally better without using balanced class weighting (Extended Data Fig. 2b).

**Separate XGBoost classifiers for each predictor category.** To determine the predictive performances of the predictor categories, we fitted an XGBoost classifier containing all the predictors from the specific category (see Supplementary Tables 2 and 3 for which predictors are included in which category). In addition, age and sex were used as predictors in each model, and a separate baseline model including age and sex only was trained to serve as a benchmark. The results from these XGBoost models are shown in Figs. 2a and 4c, and the AUCs are listed in Supplementary Table 1.

**XGBoost classifier trained with the full set of predictors.** An XGBoost model was trained using the full set of 2,997 predictors similarly to the individual-category models described above. TreeExplainer-method from the SHAP library[20] (v.0.39.0) was used to interpret the importances of individual predictors of the full XGBoost model in terms of Shapley values. Shapley values were computed by averaging over randomly chosen training samples, covering 5% of the whole training set. We used the interventional feature perturbations with a random sample of 50 individuals from the training set as the background data. The CIs for the mean absolute SHAP values were computed by bootstrapping the test set 2,000 times. The results from this model are shown in Figs. 2a and 3. Due to undersampling the vaccinated individuals and using class weights during training, the full XGBoost model is not well calibrated. We used the method proposed in ref. 19 to show that the model can be recalibrated to predict probabilities that correspond well to the actual observed probabilities.

### Lasso classifiers

The Lasso classifiers were trained in three slightly different settings: (1) separate Lasso classifiers for each predictor, (2) separate Lasso classifiers for each combination of predictor categories and (3) Lasso classifiers trained with the full set of predictors. All models were trained on the training split of the data using fivefold cross-validation to optimize the regularization strength. The models were fitted with the cv.glmnet function from the glmnet R package (v.4.1.1)[40] with the default parameter values. Balanced class weighting was used. We separately describe the three different settings for training Lasso classifiers in the following sections.

**Separate Lasso classifiers for each predictor.** To determine the predictive power of individual predictors, we fitted a Lasso logistic regression model for each predictor including age and sex. For each predictor, the following model was fit:

$$\text{vaccination status} \sim \text{age} + \text{sex} + \text{predictor}.$$

The results from these analyses were used in Fig. 2b, and the full results are listed in Supplementary Table 2.

**Separate Lasso classifiers for each combination of predictor categories.** To quantify the importance of individual predictor categories in forecasting vaccination status, we trained additional Lasso classifier models by systematically testing each possible combination of the

12 predictor categories. Not including one predictor category in the model removes all information contained only in this predictor from the training data (notice that other predictor categories can partly or even completely contain the same information as the removed category). For example, excluding all predictors in the occupation category removes from the training set all information that cannot be explained by any other predictor category. Due to the computational complexity of this experiment (requiring the training of 4,095 separate models), Lasso was used here instead of the more computationally expensive XGBoost.

To determine the predictive performance of each combination of predictor categories, we fitted a Lasso logistic regression model containing all the predictors from the specific combination of categories. In addition, age and sex were used as predictors in each model. Given a set $C$ containing all the predictors in the specific combination, the fitted model is

$$\text{vaccination status} \sim \text{age} + \text{sex} + \sum_{i \in C} \text{predictor}_i,$$

where the index $i$ runs over all predictors in category $C$. The results from these analyses were used in Fig. 4a,b.

**Lasso classifiers trained with the full set of predictors.** To determine the overall predictive performance across all predictors, we trained Lasso logistic regression models using the full set of 2,997 predictors:

$$\text{vaccination status} \sim \text{age} + \text{sex} + \sum_{i} \text{predictor}_i,$$

where the index $i$ now runs through all predictors. The results from these analyses were used only for comparison with XGBoost, which was chosen as the primary method to calculate the combined prediction model.

**Sensitivity analysis removing individuals with no data entries in the year 2019**
As a sensitivity analysis, we removed all individuals with no data entries in the year 2019 and reran the Lasso and the logistic regression analyses. Specifically, we removed each individual meeting all of the following criteria: no disease diagnoses, no medication purchases, no social benefits, no long-term care entries, no birth register entries and zero income. This ended up removing 129,089 of the total 3,192,505 individuals in the study population, indicating that we have reliable follow-up for a large majority of the study population. We considered only these data sources, because other data sources repeat the entry from the previous year if there is no new entry for the current year. Individuals with no data entries in the year 2019 were removed from the training and test sets; otherwise, the same train/test split was used. The results from this analysis are shown in Extended Data Fig. 5.

**Calculation of partial correlations between machine learning model predictions, and clustering of predictions**
To compute the similarity between the predicted probabilities for COVID-19 vaccination uptake obtained from the models trained for each predictor category, we calculated, in the test set, partial Pearson correlations between predicted probabilities from each category and visualized these as a clustered heat map (Fig. 4c). To remove the correlation between predicted probabilities that is explained by the fact that age and sex are included in each category, we used the partial_corr function from the Python library pingouin (v.0.5.2)[41], using the default parameters. Clustering of the partial correlation coefficient matrix was computed and the heat map plotted using the clustermap function from the Python library seaborn (v.0.11.2)[42], with the default parameters (method, 'average'; metric, 'euclidean').

**Analysis of genetic predictors**
We constructed the same vaccination phenotype used for FinRegistry in both FinnGen and the Estonia Biobank, except that deaths were excluded until 31 December 2019 as data were not available over the full period (total: $N_{cases}$ = 45,202, $N_{controls}$ = 374,178; FinnGen: $N_{cases}$ = 19,338, $N_{controls}$ = 254,427; Estonian Biobank: $N_{cases}$ = 25,864, $N_{controls}$ = 119,751). The GWAS was performed using REGENIE v.2.2.4 (ref. 43) for FinnGen and SAIGE v.1.0.7 (ref. 44) for the Estonian Biobank (Supplementary Methods). To test suitability for meta-analysis, we performed genetic correlations using linkage disequilibrium score regression and hapmap SNPs[24]. Quality control was performed on each set of summary statistics from FinnGen and the Estonian Biobank, restricting SNPs to have INFO score ≥ 0.8 and MAF ≥ 0.1%. The meta-analysis was performed using METAL[21]. Genetic correlations with 23 phenotypes—including educational attainment, psychiatric disorders, physical diseases (including COVID susceptibility and severity), anthropometric traits, personality traits and general lifestyle factors—were calculated using linkage disequilibrium score regression[24] (see Supplementary Table 3 for a list of summary statistics used for each phenotype).

PGSs for vaccination status were computed using PRS-CS[45]. To remove sample overlap, prior to meta-analysis with the Estonian Biobank, we first performed a GWAS in a random 70% of the FinnGen study ($N_{cases}$ = 13,555, $N_{controls}$ = 178,081). Association testing was then restricted to the remaining 30%. We trained a logistic regression model of COVID-19 vaccination where the predictors were the vaccination PGS, age and sex by training a regression in 50% of the test set, and we calculated AUC in the remaining 50%. PGSs for COVID-19 resulting in critical illness, COVID-19 resulting in hospitalization and COVID-19 susceptibility were calculated using the same method, but association with COVID-19 vaccination uptake was performed in the full sample due to the lack of sample overlap. Release 7 of the COVID-19 Host Genetics Initiative with FinnGen and 23andMe excluded (COVID-19 critical illness: $N_{cases}$ = 17,962, $N_{controls}$ = 867,353; COVID-19 hospitalization: $N_{cases}$ = 44,549, $N_{controls}$ = 2,018,071; COVID-19 susceptibility: $N_{cases}$ = 155,026, $N_{controls}$ = 2,445,292) were used as summary statistics to calculate the PGSs[15].

To understand the impact of removing COVID-19 cases on our results, we repeated all analyses including COVID-19 cases within the FinnGen sample.

To test the causal effect of COVID-19 critical illness, hospitalization and susceptibility on vaccination status, we used MR[46]. MR-Base was used to run two-sample MR[47]. For the exposures, we selected the same summary statistics used for the PGS analysis[15], whereas for the outcome, we selected the summary statistics for vaccination status from only the FinnGen sample to prevent sample overlap. We additionally tested for the causal effects of height, type 2 diabetes and BMI, as the traits had significant genetic correlations or the genome-wide significant SNPs indicated overlapping effects. As sample overlap was not an issue for these traits, the meta-analysed summary statistics for vaccination status were used to improve power. For all tests, instrumental variables were selected using a $P$ value threshold of $5 \times 10^{-8}$ and clumped using the default parameters (window size, 10,000 kb; $r^2 < 0.001$). As a sensitivity analysis, we repeated the MR for COVID-19 phenotypes using a less stringent $P$ value threshold of $5 \times 10^{-5}$.

**Ethics declarations**
FinRegistry is a collaboration project of the Finnish Institute for Health and Welfare (THL) and the Data Science Genetic Epidemiology research group at the Institute for Molecular Medicine Finland, University of Helsinki. The FinRegistry project has received the following approvals for data access from the National Institute of Health and Welfare (THL/1776/6.02.00/2019 and subsequent amendments), DVV (VRK/5722/2019-2), the Finnish Center for Pension (ETK/SUTI 22003) and Statistics Finland (TK-53-1451-19). The FinRegistry project has received IRB approval from the National Institute of Health and Welfare (Kokous 7/2019).

Patients and control participants in FinnGen provided informed consent for biobank research, on the basis of the Finnish Biobank Act. Alternatively, separate research cohorts, collected before the Finnish Biobank Act came into effect (in September 2013) and the start of FinnGen (August 2017), were collected on the basis of study-specific consents and later transferred to the Finnish biobanks after approval by Fimea (Finnish Medicines Agency), the National Supervisory Authority for Welfare and Health. Recruitment protocols followed the biobank protocols approved by Fimea. The Coordinating Ethics Committee of the Hospital District of Helsinki and Uusimaa (HUS) statement number for the FinnGen study is HUS/990/2017.

The FinnGen study is approved by the Finnish Institute for Health and Welfare (permit numbers: THL/2031/6.02.00/2017, THL/1101/5.05.00/2017, THL/341/6.02.00/2018, THL/2222/6.02.00/2018, THL/283/6.02.00/2019, THL/1721/5.05.00/2019 and THL/1524/5.05.00/2020), the Digital and Population Data Service Agency (permit numbers: VRK43431/2017-3, VRK/6909/2018-3 and VRK/4415/2019-3), the Social Insurance Institution (permit numbers: KELA 58/522/2017, KELA 131/522/2018, KELA 70/522/2019, KELA 98/522/2019, KELA 134/522/2019, KELA 138/522/2019, KELA 2/522/2020 and KELA 16/522/2020), Findata (permit numbers: THL/2364/14.02/2020, THL/4055/14.06.00/2020, THL/3433/14.06.00/2020, THL/4432/14.06.00/2020, THL/5189/14.06/2020, THL/5894/14.06.00/2020, THL/6619/14.06.00/2020, THL/209/14.06.00/2021, THL/688/14.06.00/2021, THL/1284/14.06.00/2021, THL/1965/14.06.00/2021, THL/5546/14.02.00/2020, THL/2658/14.06.00/2021 and THL/4235/14.06.00/202), Statistics Finland (permit numbers: TK-53-1041-17, TK/143/07.03.00/2020 (earlier TK-53-90-20), TK/1735/07.03.00/2021 and TK/3112/07.03.00/2021) and the Finnish Registry for Kidney Diseases (permission/extract from the meeting minutes on 4 July 2019).

The Biobank Access Decisions for FinnGen samples and data utilized in FinnGen Data Freeze 9 include THL Biobank BB2017_55, BB2017_111, BB2018_19, BB_2018_34, BB_2018_67, BB_2018_71, BB2019_7, BB2019_8, BB2019_26, BB2020_1, Finnish Red Cross Blood Service Biobank 7 December 2017, Helsinki Biobank HUS/359/2017, HUS/248/2020, Auria Biobank AB17-5154 and amendment no. 1 (17 August 2020), AB20-5926 and amendment no. 1 (23 April 2020) and its modification (22 September 2021), Biobank Borealis of Northern Finland_2017_1013, Biobank of Eastern Finland 1186/2018 and amendment 22 § /2020, Finnish Clinical Biobank Tampere MH0004 and amendments (21 February 2020 and 6 October 2020), Central Finland Biobank 1-2017, and Terveystalo Biobank STB 2018001 and amendment 25 August 2020.

The activities of the Estonian Biobank are regulated by the Human Genes Research Act, which was adopted in 2000 specifically for the operations of the Estonian Biobank. Analysis of individual-level data from the Estonian Biobank was carried out under ethical approval no. 1.1-12/3022 from the Estonian Committee on Bioethics and Human Research (Estonian Ministry of Social Affairs), and according to data release application 3-10/GI/31487 from the Estonian Biobank, Institute of Genomics, University of Tartu.

### Reporting summary

Further information on research design is available in the Nature Portfolio Reporting Summary linked to this article.

## Data availability

Data dictionaries for FinRegistry are publicly available on the FinRegistry website (www.finregistry.fi/finnish-registry-data). Access to the FinRegistry data can be obtained by submitting a data permit application for individual-level data to the Finnish social and health data permit authority, Findata (https://asiointi.findata.fi/). The application includes information on the purpose of data use; the requested data, including the variables, definitions of the target and control groups, and external datasets to be combined with FinRegistry data; the dates of the data needed; and a data utilization plan. The requests are evaluated case by case. Once approved, the data are sent to a secure computing environment (Kapseli) and can be accessed within the European Economic Area and within countries with an adequacy decision from the European Commission. The Finnish biobank data can be accessed through the Fingenious services (https://site.fingenious.fi/en/) managed by FINBB. Access to the Estonian Biobank data (https://genomics.ut.ee/en/content/estonian-biobank) is restricted to approved researchers and can be requested. Summary statistics of the COVID-19 vaccination uptake GWAS are available in the GWAS catalogue under accession code GCST90255613.

## Code availability

The essential analysis code used to produce the results is available in the FinRegistry GitHub at https://github.com/dsgelab/COVID-19-vaccination-public.

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

## Acknowledgements

We thank the entire FinRegistry, FinnGen and Estonia Biobank teams for making the data available for the study, and we acknowledge CSC—IT Center for Science, Finland, for computational resources. This study has received funding from the European Union's Horizon 2020 research and innovation programme under grant agreement no. 101016775. The FinRegistry project has received funding from the European Research Council under the European Union's Horizon 2020 research and innovation programme (grant agreement no. 945733), starting grant AI-Prevent. This Estonian Biobank study was funded by the European Union through the European Regional Development Fund Project No. 2014-2020.4.01.15-0012 GENTRANSMED. Data analysis was carried out in part in the High-Performance Computing Center of the University of Tartu. We acknowledge the participants and investigators in the FinnGen study. The FinnGen project is funded by two grants from Business Finland (HUS 4685/31/2016 and UH 4386/31/2016) and the following industry partners: AbbVie Inc., AstraZeneca UK Ltd, Biogen MA Inc., Bristol Myers Squibb (and Celgene Corporation & Celgene International II Sàrl), Genentech Inc., Merck Sharp & Dohme LCC, Pfizer Inc., GlaxoSmithKline Intellectual Property Development Ltd, Sanofi US Services Inc., Maze Therapeutics Inc., Janssen Biotech Inc., Novartis AG, and Boehringer Ingelheim International GmbH. We acknowledge the following biobanks for delivering biobank samples to FinnGen: Auria Biobank (www.auria.fi/biopankki), THL Biobank (www.thl.fi/biobank), Helsinki Biobank (www.helsinginbiopankki.fi), Biobank Borealis of Northern Finland (https://www.ppshp.fi/Tutkimus-ja-opetus/Biopankki/Pages/Biobank-Borealis-briefly-in-English.aspx), Finnish Clinical Biobank Tampere (www.tays.fi/en-US/Research_and_development/Finnish_Clinical_Biobank_Tampere), Biobank of Eastern Finland (www.ita-suomenbiopankki.fi/en), Central Finland Biobank (www.ksshp.fi/fi-FI/Potilaalle/Biopankki), Finnish Red Cross Blood Service Biobank (www.veripalvelu.fi/verenluovutus/biopankkitoiminta), Terveystalo Biobank (www.terveystalo.com/fi/Yritystietoa/Terveystalo-Biopankki/Biopankki/) and Arctic Biobank (https://www.oulu.fi/en/university/faculties-and-units/faculty-medicine/northern-finland-birth-cohorts-and-arctic-biobank). All Finnish biobanks are members of BBMRI.fi infrastructure (www.bbmri.

fi). Finnish Biobank Cooperative (FINBB, https://finbb.fi/) is the coordinator of BBMRI-ERIC operations in Finland. The funders had no role in study design, data collection and analysis, decision to publish or preparation of the manuscript.

## Author contributions

T.H., A.G., B.J. and P.V. wrote the manuscript with input and comments from H.S., K.K., A.V., T.L., H.N., J.S., R.M., M.D., H.M.O., L.M., M.P. and S.R. T.H. performed all analyses using the nationwide FinRegistry dataset. B.J. performed all genetic analyses using FinnGen and all genetic meta-analyses. H.S. and K.K. performed the genetic analyses using Estonian Biobank. T.H., B.J., A.V., P.V. and A.G. preprocessed and curated the data for the nationwide FinRegistry analyses. A.G., M.P., S.R., R.M., L.M. and H.M.O. supervised the study.

## Funding

## Competing interests

The authors declare no competing interests.

## Additional information

**Extended data** is available for this paper at https://doi.org/10.1038/s41562-023-01591-z.

**Correspondence and requests for materials** should be addressed to Andrea Ganna.

### FinnGen

**Mark Daly**[1,3,4,5], **Hanna M. Ollila**[1,3,6,7], **Markus Perola**[8], **Samuli Ripatti**[1,3,4,9] **& Andrea Ganna**[1,3,4]

A full list of members and their affiliations appears in the Supplementary Information.

### Estonian Biobank Research Team

**Andres Metspalu**[2], **Tõnu Esko**[2], **Reedik Mägi**[2], **Mari Nelis**[2] **& Georgi Hudjashov**[2]

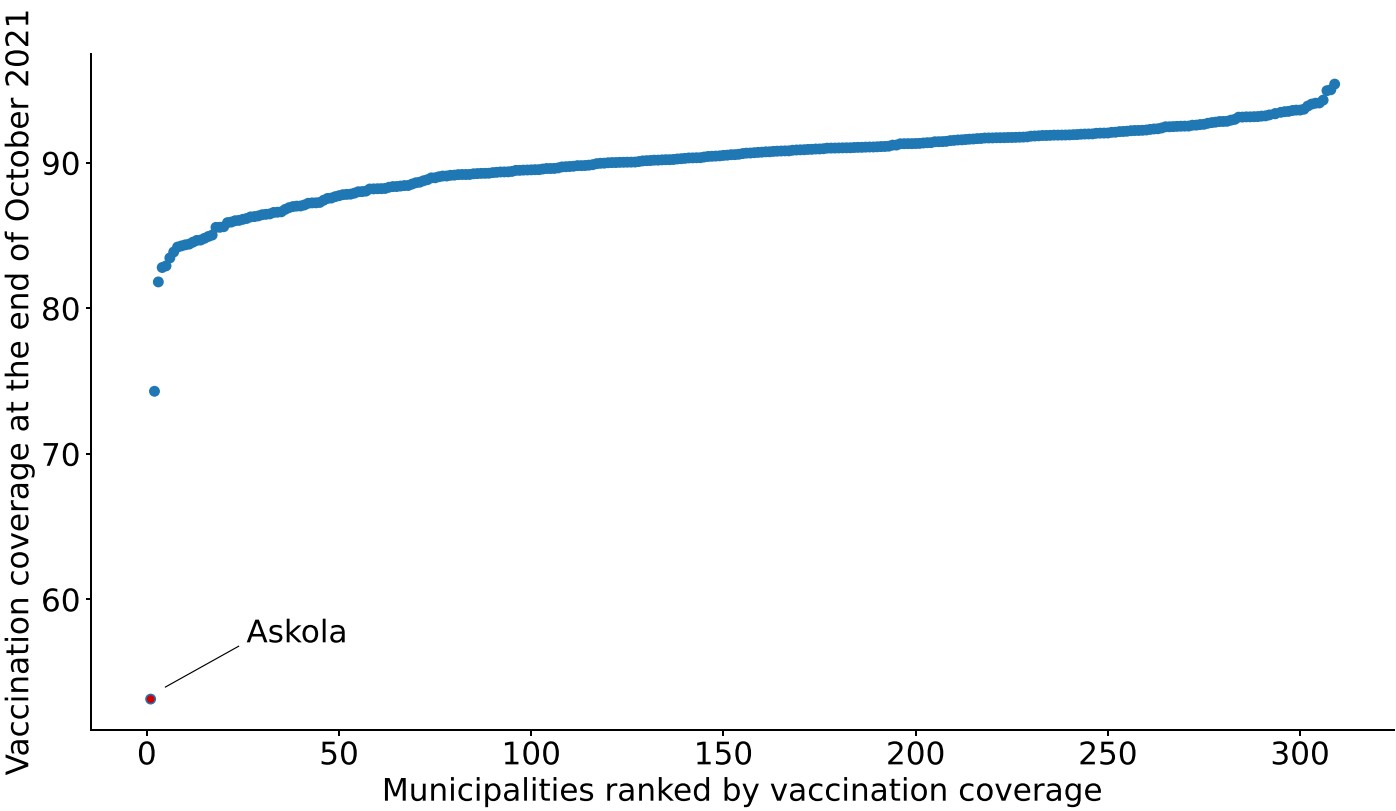

**Extended Data Fig. 1 | COVID-19 1st dose vaccination coverage in the study population in each Finnish municipality.** Residents of Askola (highlighted with red and annotated) were excluded from the study as the vaccination coverage in Askola (2,948 residents in the study population) seemed artificially low compared to all other municipalities and is likely due to misreporting.

**a**

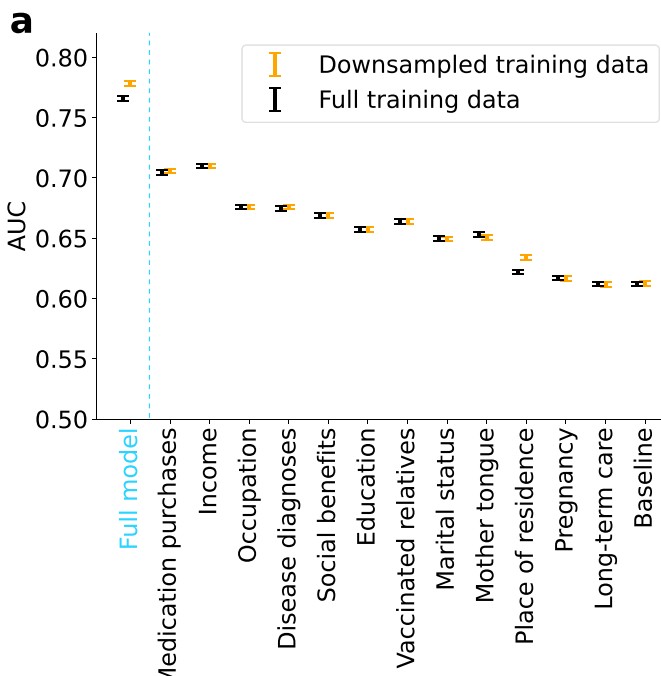

**b**

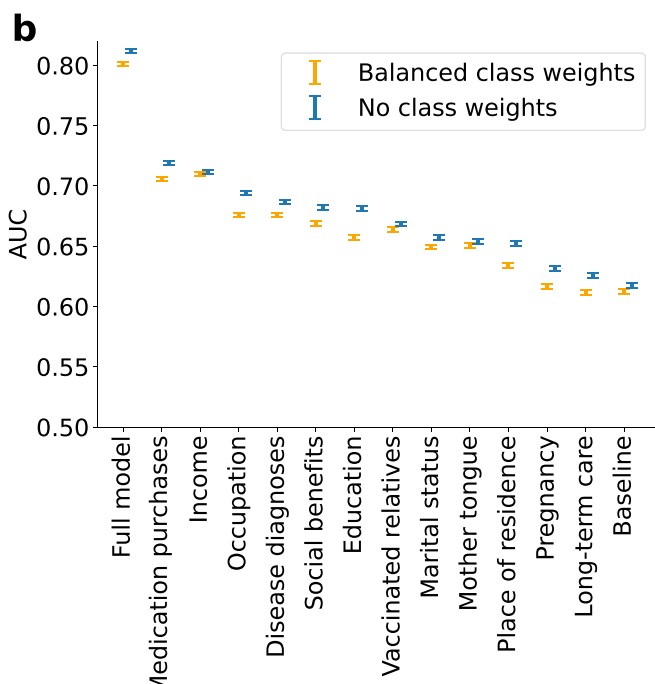

**Extended Data Fig. 2 | The effects of downsampling of controls and use of balanced class weights to the XGBoost model predictions. a)** Downsampling controls does not negatively affect the machine learning model predictions. AUCs for models trained with all cases and five randomly sampled controls per each case (orange) and for models trained with the full training data without downsampling (black). Predictors used by the models are indicated on the x-axis. All AUCs correspond to XGBoost models, except for the Full model (indicated with blue colour), where the AUCs were computed for the Lasso model, as the full XGBoost model could not be trained without downsampling the controls due to

memory issues. **b)** Class weighting has a negligible effect on the XGBoost model predictions. AUCs for XGBoost models trained with balanced class weighting (orange) versus with no class weights (blue). In both cases, five controls per each case were sampled randomly for the training data. Predictors used by the models are indicated on the x-axis. In both panels, the error bars indicate 95% confidence intervals computed using bootstrapping, and the centre of the error bars corresponds to the point estimate. All models include the baseline predictors age and sex.

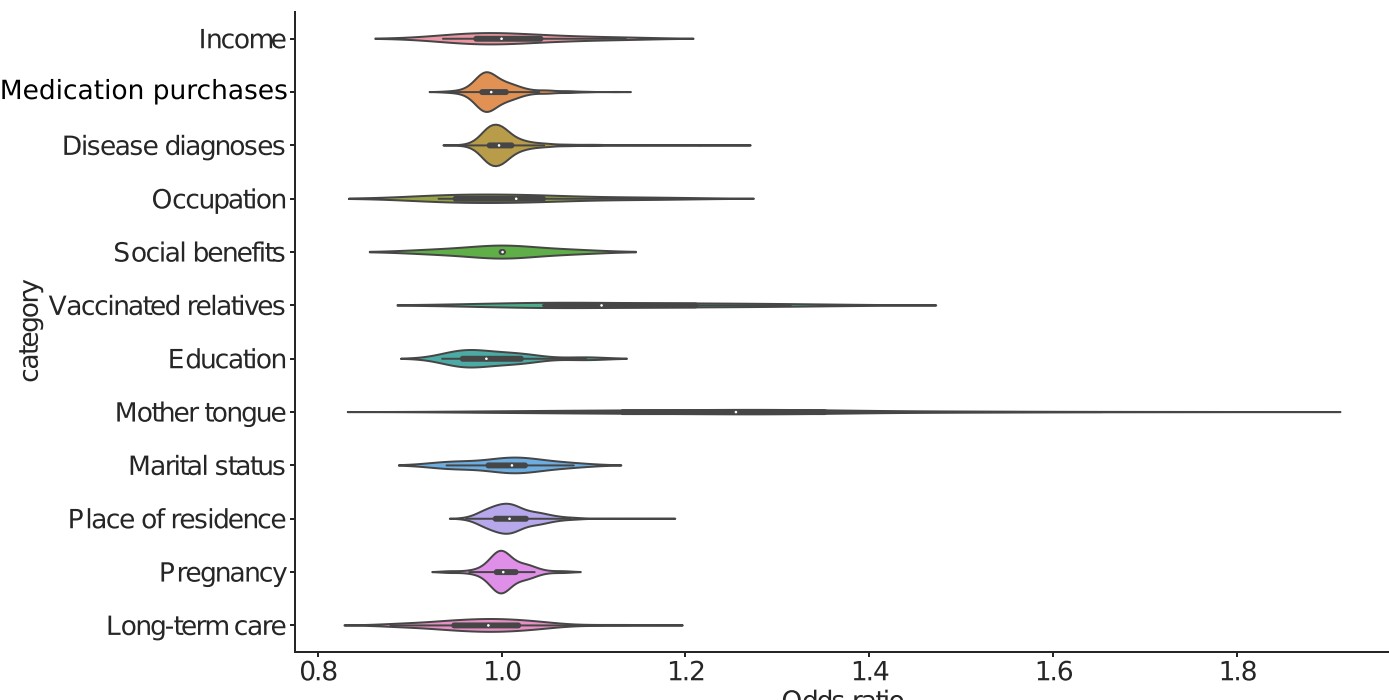

**Extended Data Fig. 3 | Effect size distributions across the predictor categories.** Violin plots describing the distributions of adjusted odds ratios (OR) (adjusted for age and sex, see Methods) for not uptaking the COVID-19 vaccination separately for each of the predictor categories. See Supplementary Table 3 for a full list of ORs for the individual predictors. Inside the violins, the box shows the quartiles of the distribution, white dot is the median and whiskers correspond to 1.5 times the interquartile range.

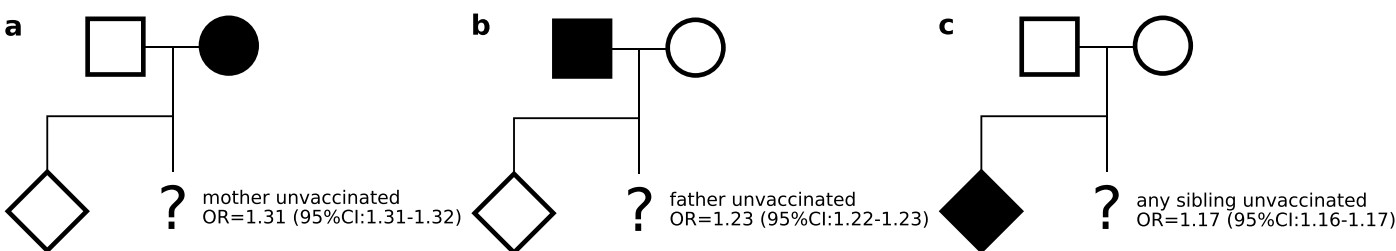

**a** mother unvaccinated
OR=1.31 (95%CI:1.31-1.32)

**b** father unvaccinated
OR=1.23 (95%CI:1.22-1.23)

**c** any sibling unvaccinated
OR=1.17 (95%CI:1.16-1.17)

**Extended Data Fig. 4 | The effect of unvaccinated relatives to risk of not vaccinating.** Adjusted (for age and sex, see Methods) odds ratios (OR) describing the risk of not uptaking the COVID-19 vaccination when either **a**) mother, **b**) father, or **c**) any of their siblings is unvaccinated (for the entire follow-up period of 1.1.2021-31.10.2021).

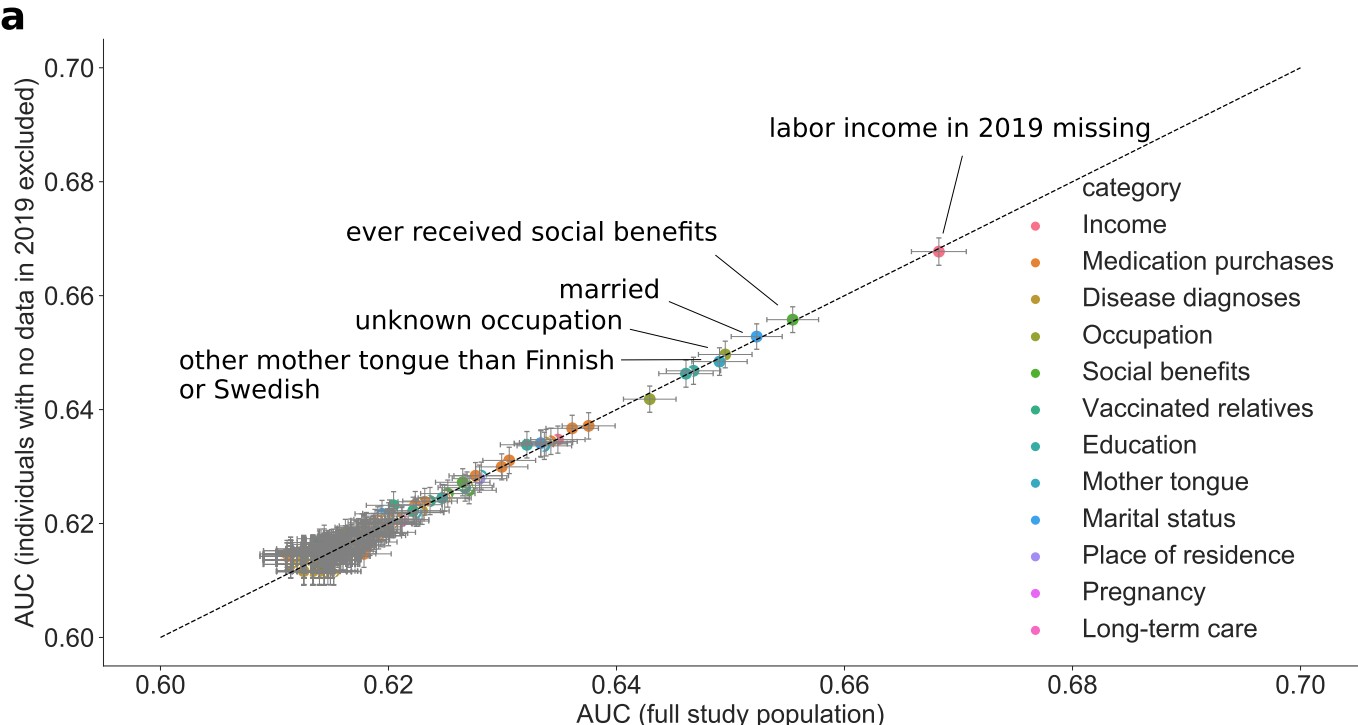

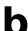

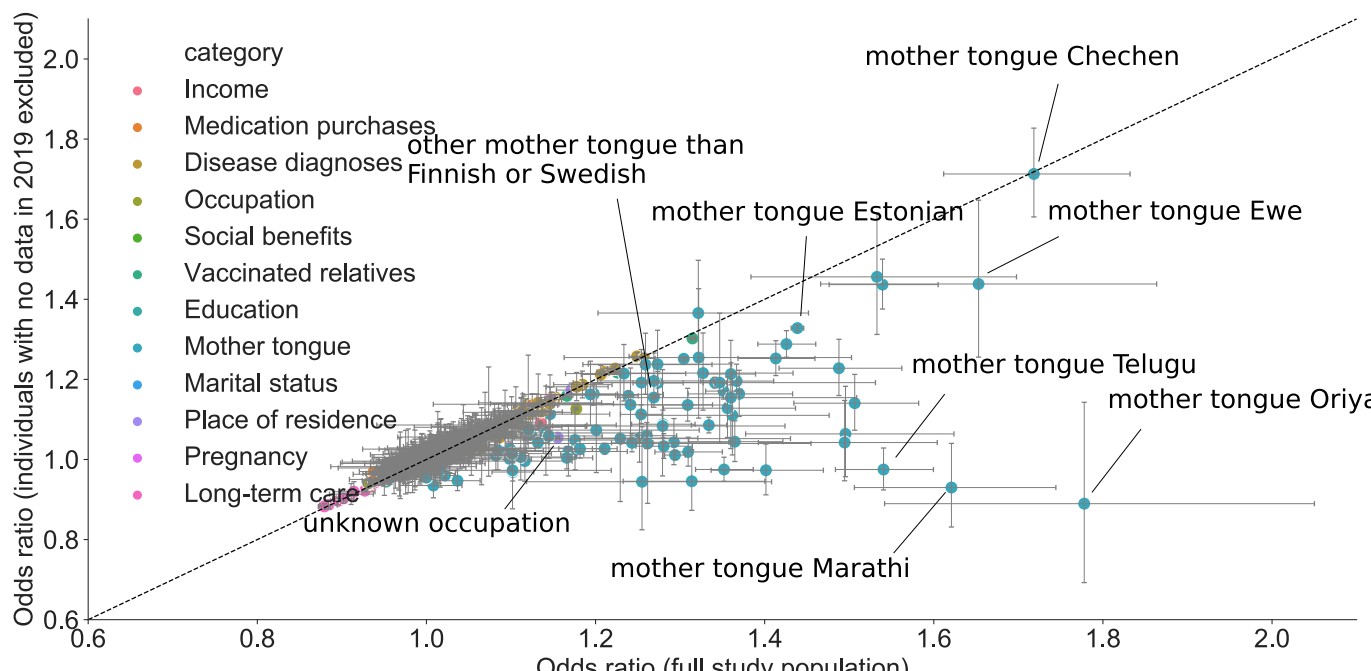

**Extended Data Fig. 5 | Sensitivity analysis removing all individuals with no data entries in the year 2019 from the study population. In total 129,089 individuals had no data entries in the year 2019 (see Methods for details). The dots are coloured by the predictor category. Error bars correspond to 95% confidence intervals computed using bootstrapping and dots correspond to point estimates. a)** Area under receiver-operator characteristics curve (AUC) using the full study population (x-axis) plotted against the AUC using the study population with individuals with no data in the year 2019 removed (y-axis) from Lasso classifier models trained separately for each individual predictor (including also age and sex as predictors in each model). Models were trained separately using training data with and without individuals with no data entries in the year 2019. AUCs were computed on a separate unseen test set. No significant changes in AUC were observed for any predictor. **b)** Odds ratios (OR) using the full study population (x-axis) plotted against the ORs using the study population with individuals with no data in the year 2019 removed (y-axis) from logistic regression models trained separately for each individual predictor, adjusting for age and sex. Significant drop in OR when removing individuals with no data in the year 2019 occur mostly for relatively rare mother tongues (some highlighted with labels).

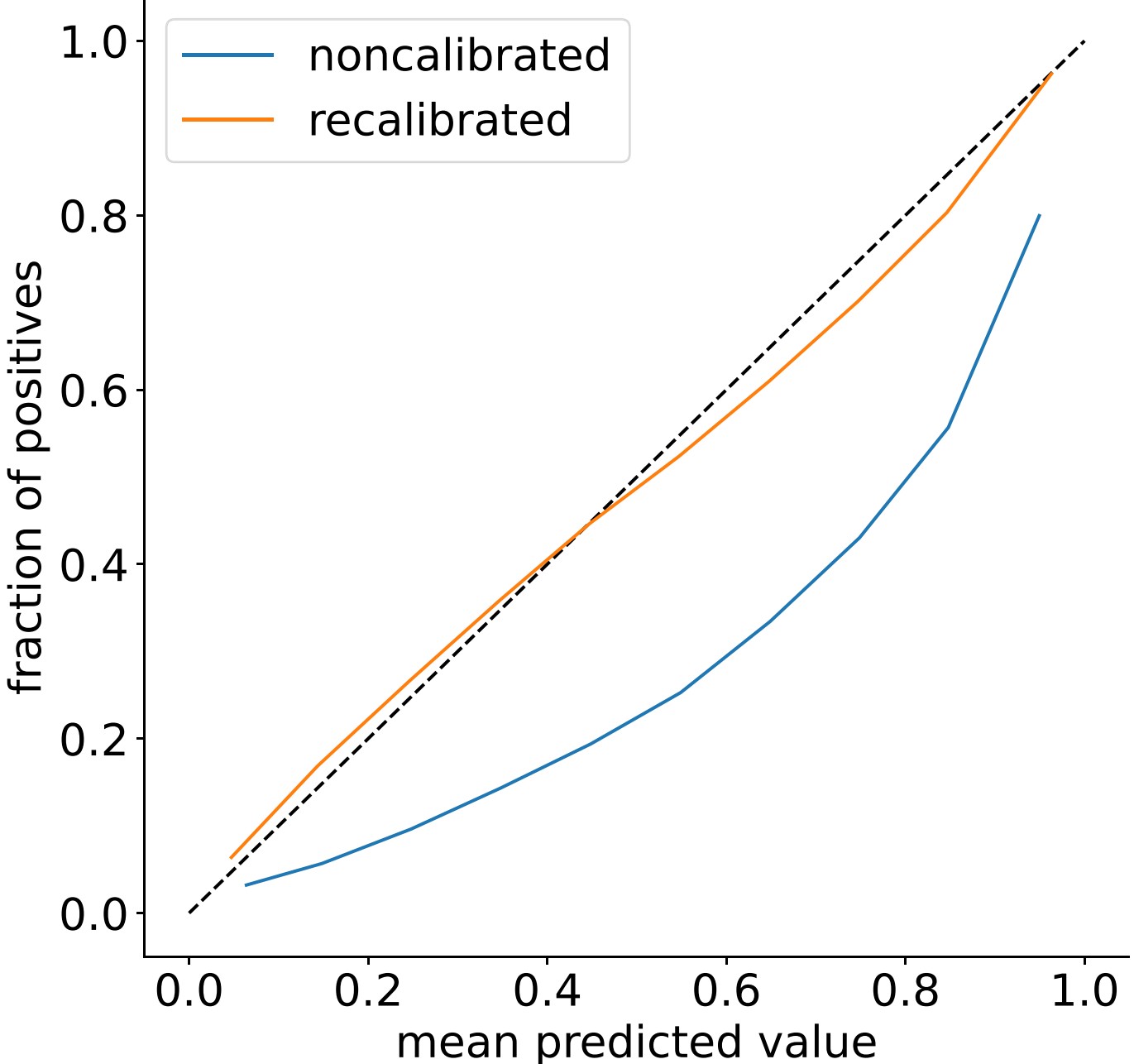

**Extended Data Fig. 6 | Calibration of the prediction model of COVID-19 vaccination uptake.** Calibration curves for the full XGBoost (all predictors) model predicting COVID-19 vaccination status before (blue) and after (orange) recalibration (see Methods).

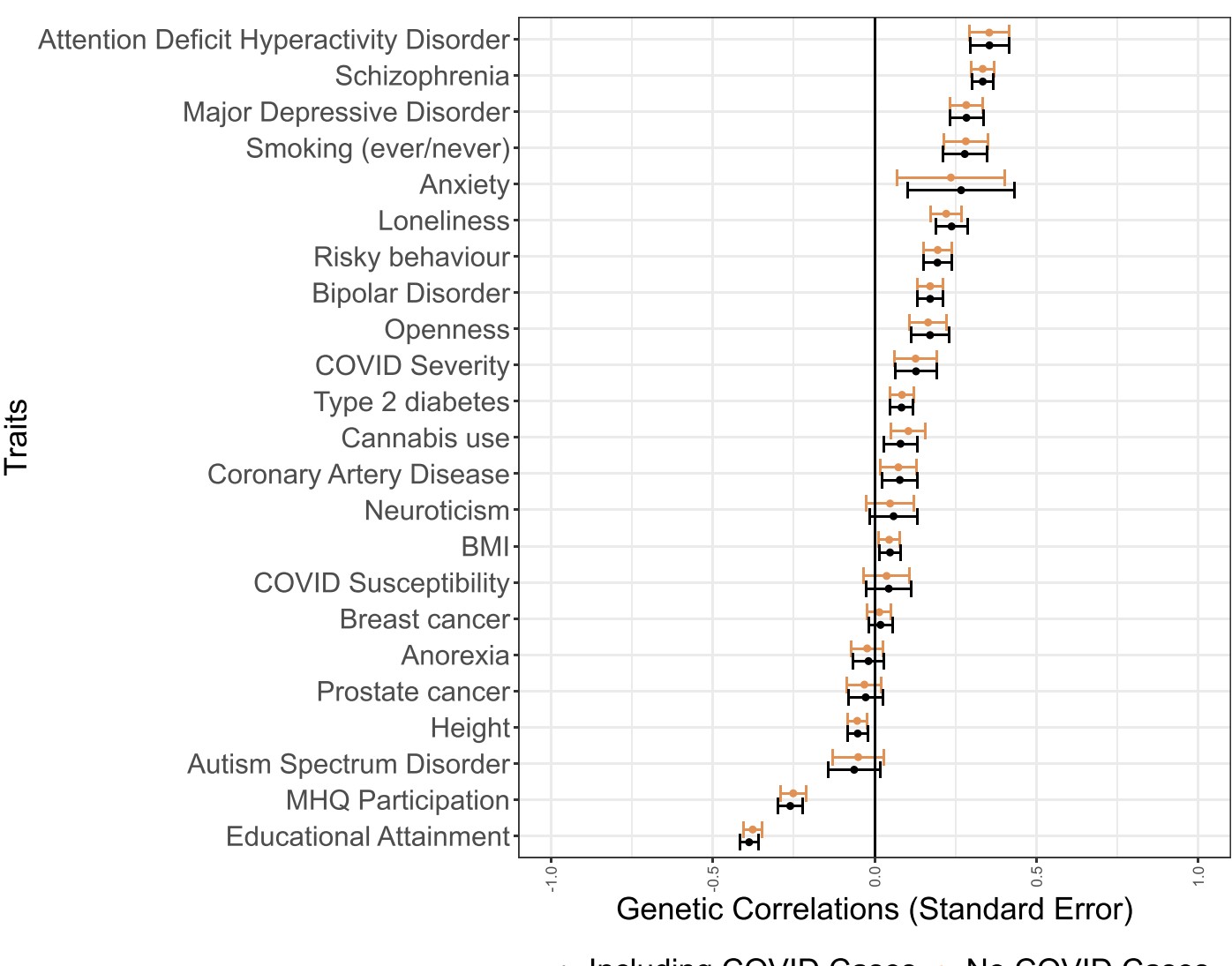

**Extended Data Fig. 7 | Genetic correlations with and without COVID-19 cases included in the phenotype definition.** The analysis was performed within the FinnGen study. Point estimates represent correlations with error bars reflecting standard errors. Black error bars and point estimates represent the vaccination phenotype which includes COVID-19 cases.

# Reporting Summary

## Statistics

For all statistical analyses, confirm that the following items are present in the figure legend, table legend, main text, or Methods section.

| n/a | Confirmed | |
|---|---|---|
| ☐ | ☒ | The exact sample size (*n*) for each experimental group/condition, given as a discrete number and unit of measurement |
| ☐ | ☒ | A statement on whether measurements were taken from distinct samples or whether the same sample was measured repeatedly |
| ☐ | ☒ | The statistical test(s) used AND whether they are one- or two-sided<br>*Only common tests should be described solely by name; describe more complex techniques in the Methods section.* |
| ☐ | ☒ | A description of all covariates tested |
| ☐ | ☒ | A description of any assumptions or corrections, such as tests of normality and adjustment for multiple comparisons |
| ☐ | ☒ | A full description of the statistical parameters including central tendency (e.g. means) or other basic estimates (e.g. regression coefficient) AND variation (e.g. standard deviation) or associated estimates of uncertainty (e.g. confidence intervals) |
| ☐ | ☒ | For null hypothesis testing, the test statistic (e.g. *F*, *t*, *r*) with confidence intervals, effect sizes, degrees of freedom and *P* value noted<br>*Give P values as exact values whenever suitable.* |
| ☒ | ☐ | For Bayesian analysis, information on the choice of priors and Markov chain Monte Carlo settings |
| ☒ | ☐ | For hierarchical and complex designs, identification of the appropriate level for tests and full reporting of outcomes |
| ☐ | ☒ | Estimates of effect sizes (e.g. Cohen's *d*, Pearson's *r*), indicating how they were calculated |

*Our web collection on statistics for biologists contains articles on many of the points above.*

## Software and code

Policy information about availability of computer code

| | |
|---|---|
| Data collection | No software was used for data collection. |
| Data analysis | All data analysis steps are described in Methods. Briefly, logistic regression analyses were performed with R using bigglm from library biglm (version 0.9.2.1), and multiple hypothesis testing corrected p-values were computed using Python package statsmodels (version 0.12.2). XGBoost analyses were performed with the Python library xgboost (version 1.5.0), with hyperparameters optimized using Bayesian hyperparameter optimization (BayesSearchCV function from scikit-optimize, version 0.9.0). XGBoost model feature importances were computed using the shap software (version 0.39.0). Lasso classifiers were fitted with the cv.glmnet function from the glmnet R package (version 4.1.1). Partial correlations were computed using partial_corr function from Python library pingouin (version 0.5.2). Clustering of the partial correlation coefficient matrix was computed and the heatmap plotted using the clustermap function from Python library seaborn (version 0.11.2). GWAS was performed using REGENIE v2.2.4 for FinnGen and SAIGE v1.0.7 for Estonian Biobank. Genetic correlations were computed using Linkage Disequilibrium Score Regression (LDSC, version 1.0.1). Meta-analysis was performed using METAL (version released on 2011-03-25). Polygenic Scores (PGS) were computed using PRS-CS (version released on 2022-11-03). MRBase (TwoSampleMR, version 0.5.6) was used to run two sample mendelian randomization. The most likely gene linked to each lead variant in the GWAS was reported based on a machine learning-based prioritization score from Open Targets Genetics. The essential custom code developed for this study is available in Github at: https://github.com/dsgelab/COVID-19-vaccination-public. |

For manuscripts utilizing custom algorithms or software that are central to the research but not yet described in published literature, software must be made available to editors and reviewers. We strongly encourage code deposition in a community repository (e.g. GitHub). See the Nature Portfolio guidelines for submitting code & software for further information.

# Data

Policy information about availability of data

All manuscripts must include a data availability statement. This statement should provide the following information, where applicable:
- Accession codes, unique identifiers, or web links for publicly available datasets
- A description of any restrictions on data availability
- For clinical datasets or third party data, please ensure that the statement adheres to our policy

Data dictionaries for FinRegistry are publicly available on the FinRegistry website (www.finregistry.fi/finnish-registry-data). Access to FinRegistry data can be obtained by submitting a data permit application for individual-level data for the Finnish social and health data permit authority Findata (https://asiointi.findata.fi/). The application includes information on the purpose of data use; the requested data, including the variables, definitions for the target and control groups, and external datasets to be combined with FinRegistry data; the dates of the data needed; and a data utilization plan. The requests are evaluated on a case-by-case basis. Once approved, the data are sent to a secure computing environment Kapseli and can be accessed within the European Economic Area (EEA) and within countries with an adequacy decision from the European Commission.

The Finnish biobank data can be accessed through the Fingenious® services (https://site.fingenious.fi/en/) managed by FINBB.

Access to Estonian biobank data (https://genomics.ut.ee/en/content/estonian-biobank) is restricted for approved researchers and can be requested.

Summary statistics of the COVID-19 vaccination uptake GWAS are available at the GWAS catalog with accession code GCP000553.

# Human research participants

Policy information about studies involving human research participants and Sex and Gender in Research.

| | |
|---|---|
| Reporting on sex and gender | Sex (as recorded in the nation-wide registers) was used as a covariate in all of the presented analyses. |
| Population characteristics | The FinRegistry dataset includes all individuals alive and living in Finland on 31.10.2021, aged between 30-80. Individuals having received a laboratory confirmed COVID-19 diagnosis prior to 31.10.2021, or who lived in one municipality with insufficient vaccination records were further excluded. After these exclusion criteria, the final FinRegistry study population contains 3,192,505 individuals. The FinnGen study population includes 273,615 of these individuals with genetic information measured. Similar inclusion and exclusion criteria were applied to the Estonian biobank, resulting in 202,910 individuals with genetic information measured. |
| Recruitment | The FinRegistry dataset includes each individual alive and living in Finland on 1.1.2010. |
| Ethics oversight | FinRegistry is a collaboration project of the Finnish Institute for Health and Welfare (THL) and the Data Science Genetic Epidemiology research group at the Institute for Molecular Medicine Finland (FIMM), University of Helsinki. The FinRegistry project has received the following approvals for data access from the National Institute of Health and Welfare (THL/1776/6.02.00/2019 and subsequent amendments), DVV (VRK/5722/2019-2), Finnish Center for Pension (ETK/SUTI 22003) and Statistics Finland (TK-53-1451-19). The FinRegistry project has received IRB approval from the National Institute of Health and Welfare (Kokous 7/2019).<br><br>Patients and control subjects in FinnGen provided informed consent for biobank research, based on the Finnish Biobank Act. Alternatively, separate research cohorts, collected prior the Finnish Biobank Act came into effect (in September 2013) and start of FinnGen (August 2017), were collected based on study-specific consents and later transferred to the Finnish biobanks after approval by Fimea (Finnish Medicines Agency), the National Supervisory Authority for Welfare and Health. Recruitment protocols followed the biobank protocols approved by Fimea. The Coordinating Ethics Committee of the Hospital District of Helsinki and Uusimaa (HUS) statement number for the FinnGen study is Nr HUS/990/2017.<br>The FinnGen study is approved by Finnish Institute for Health and Welfare (permit numbers: THL/2031/6.02.00/2017, THL/1101/5.05.00/2017, THL/341/6.02.00/2018, THL/2222/6.02.00/2018, THL/283/6.02.00/2019, THL/1721/5.05.00/2019 and THL/1524/5.05.00/2020), Digital and population data service agency (permit numbers: VRK43431/2017-3, VRK/6909/2018-3, VRK/4415/2019-3), the Social Insurance Institution (permit numbers: KELA 58/522/2017, KELA 131/522/2018, KELA 70/522/2019, KELA 98/522/2019, KELA 134/522/2019, KELA 138/522/2019, KELA 2/522/2020, KELA 16/522/2020), Findata permit numbers THL/2364/14.02/2020, THL/4055/14.06.00/2020,,THL/3433/14.06.00/2020, THL/4432/14.06/2020, THL/5189/14.06/2020, THL/5894/14.06.00/2020, THL/6619/14.06.00/2020, THL/209/14.06.00/2021, THL/688/14.06.00/2021, THL/1284/14.06.00/2021, THL/1965/14.06.00/2021, THL/5546/14.02.00/2020, THL/2658/14.06.00/2021, THL/4235/14.06.00/202, Statistics Finland (permit numbers: TK-53-1041-17 and TK/143/07.03.00/2020 (earlier TK-53-90-20) TK/1735/07.03.00/2021, TK/3112/07.03.00/2021) and Finnish Registry for Kidney Diseases permission/extract from the meeting minutes on 4th July 2019.<br>The Biobank Access Decisions for FinnGen samples and data utilized in FinnGen Data Freeze 9 include: THL Biobank BB2017_55, BB2017_111, BB2018_19, BB_2018_34, BB_2018_67, BB2018_71, BB2019_7, BB2019_8, BB2019_26, BB2020_1, Finnish Red Cross Blood Service Biobank 7.12.2017, Helsinki Biobank HUS/359/2017, HUS/248/2020, Auria Biobank AB17-5154 and amendment #1 (August 17 2020), AB20-5926 and amendment #1 (April 23 2020) and it´s modification (Sep 22 2021), Biobank Borealis of Northern Finland_2017_1013, Biobank of Eastern Finland 1186/2018 and amendment 22 § /2020, Finnish Clinical Biobank Tampere MH0004 and amendments (21.02.2020 & 06.10.2020), Central Finland Biobank 1-2017, and Terveystalo Biobank STB 2018001 and amendment 25th Aug 2020.<br><br>The activities of the Estonian Biobank (EstBB) are regulated by the Human Genes Research Act, which was adopted in 2000 |

specifically for the operations of the EstBB. Analysis of individual level data from the EstBB was carried out under ethical approval 1.1-12/3022 from the Estonian Committee on Bioethics and Human Research (Estonian Ministry of Social Affairs), and according to data release application 3-10/GI/31487 from the Estonian Biobank, Institute of Genomics, University of Tartu.

Note that full information on the approval of the study protocol must also be provided in the manuscript.

# Field-specific reporting

Please select the one below that is the best fit for your research. If you are not sure, read the appropriate sections before making your selection.

☒ Life sciences ☐ Behavioural & social sciences ☐ Ecological, evolutionary & environmental sciences

For a reference copy of the document with all sections, see nature.com/documents/nr-reporting-summary-flat.pdf

# Life sciences study design

All studies must disclose on these points even when the disclosure is negative.

| Sample size | The FinRegistry dataset includes each individual alive and living in Finland on 1.1.2010, which means the entire population of Finland aged 30-80 on 31.10.2021 is included in the study. From FinnGen and Estonia biobank, all genotyped individuals meeting the inclusion criteria were selected. |
| --- | --- |
| Data exclusions | To restrict the study population to individuals who had had a fair opportunity of receiving the first dose of a COVID-19 vaccination by the end of October 2021, we excluded the following individuals:<br><br>Individuals who had died or emigrated before 31.12.2020 (death statistics for year 2021 in Finland were not available).<br>Individuals who were less than 30 years old at 31.10.2021.<br>Individuals who were older than 80 years old at 31.10.2021.<br>Individuals who had a laboratory-confirmed COVID-19 diagnosis prior to 31.10.2021.<br>Individuals living in a municipality called Askola.<br><br>For the genetic analyses conducted in FinnGen and Estonian Biobank, death or emigration was limited to 31.12.2019 as statistics beyond this date were unavailable for FinnGen. Residents of Askola were excluded, as it was the only municipality where the vaccination coverage differed radically from any other Finnish municipality. |
| Replication | The GWAS of vaccination uptake was first performed in the FinnGen sample, and replicating the analysis in one independent cohort, the Estonian biobank. The genetic correlation between these analysis was high (0.8, 95% CI: 0.66-0.95). |
| Randomization | Not relevant, as division to cases and controls was made according to the vaccination status and a nationwide sample was used. |
| Blinding | Analyses were performed using computational algorithms for large data sets of registry and genetic data and blinding was thus not relevant for this study. |

# Reporting for specific materials, systems and methods

We require information from authors about some types of materials, experimental systems and methods used in many studies. Here, indicate whether each material, system or method listed is relevant to your study. If you are not sure if a list item applies to your research, read the appropriate section before selecting a response.

## Materials & experimental systems

| n/a | Involved in the study |
| --- | --- |
| ☒ | Antibodies |
| ☒ | Eukaryotic cell lines |
| ☒ | Palaeontology and archaeology |
| ☒ | Animals and other organisms |
| ☒ | Clinical data |
| ☒ | Dual use research of concern |

## Methods

| n/a | Involved in the study |
| --- | --- |
| ☒ | ChIP-seq |
| ☒ | Flow cytometry |
| ☒ | MRI-based neuroimaging |

