## [Peer Review File · Nature Human Behaviour]

Peer Review Information

Journal: Nature Human Behaviour

Manuscript Title: Nationwide health, socioeconomic, and genetic predictors of COVID-19 vaccination status in Finland

Corresponding author name(s): Andrea Ganna

Editorial Notes:

Redactions – unpublished data	Parts of this Peer Review File have been redacted as indicated to maintain the confidentiality of unpublished data.
Redactions – confidential patient information	Parts of this Peer Review File have been redacted as indicated to maintain patient confidentiality.
Redactions – published data	Parts of this Peer Review File have been redacted as indicated to remove third-party material.

Reviewer Comments & Decisions:

Decision Letter, initial version:

19th December 2022

Dear Dr Ganna,

Thank you once again for your manuscript, entitled "Health, socioeconomic and genetic predictors of COVID-19 vaccination uptake: a nationwide machine-learning study," and for your patience during the peer review process.

Your manuscript has now been evaluated by 3 reviewers, whose comments are included at the end of this letter. Although the reviewers find your work to be of interest, they also raise some important concerns. We are interested in the possibility of publishing your study in Nature Human Behaviour, but would like to consider your response to these concerns in the form of a revised manuscript before we make a decision on publication.

In your revision, please follow Reviewer #3's advice on including additional Mendelian Randomization analyses to strengthen your causal claims. Please also provide additional discussion of limitations and implications arising from your work, and include more detailed explanations of your methodological choices. Finally, Reviewer #4 has a number of concerns regarding your ML methodological choices and we ask that you address them and provide the clarifications.

Finally, your revised manuscript must comply fully with our editorial policies and formatting requirements. Failure to do so will result in your manuscript being returned to you, which will delay its consideration. To assist you in this process, I have attached a checklist that lists all of our

requirements. If you have any questions about any of our policies or formatting, please don't hesitate to contact me.

In sum, we invite you to revise your manuscript taking into account all reviewer and editor comments. We are committed to providing a fair and constructive peer-review process. Do not hesitate to contact us if there are specific requests from the reviewers that you believe are technically impossible or unlikely to yield a meaningful outcome.

We hope to receive your revised manuscript within two months. I would be grateful if you could contact us as soon as possible if you foresee difficulties with meeting this target resubmission date.

- Include a "Response to the editors and reviewers" document detailing, point-by-point, how you addressed each editor and referee comment. If no action was taken to address a point, you must provide a compelling argument. When formatting this document, please respond to each reviewer comment individually, including the full text of the reviewer comment verbatim followed by your response to the individual point. This response will be used by the editors to evaluate your revision and sent back to the reviewers along with the revised manuscript.
- Highlight all changes made to your manuscript or provide us with a version that tracks changes.

[REDACTED]

We look forward to seeing the revised manuscript and thank you for the opportunity to review your work. Please do not hesitate to contact me if you have any questions or would like to discuss these revisions further.

Sincerely,

Arunas Radzvilavicius, PhD
Editor, Nature Human Behaviour
Nature Research

Reviewer expertise:

Reviewer #1: vaccine hesitancy, conspiracy theories

Reviewer #2: public health

Reviewer #3: genetics and behavioural genomics

Reviewer #4: machine learning

REVIEWER COMMENTS:

Reviewer #1:

Remarks to the Author:

This is a fascinating paper using great data. Some minor comments.

1. It seems like you are saying that the genetic indicators are both strongly and weakly related. Some of the discussion comes of as inconsistent.
2. The criticisms of the survey works are unnecessary. Maybe you want to point to specific papers instead of making these blanket criticisms on pages 81-95. The good thing about surveys is that many of them, particularly for COVID vaxx choices, have been fairly comprehensive. And, I am not sure that your data fully overcomes the sampling bias problems.
3. One concern I have is that many of your predictor variables are themselves outcomes, no different than COVID vaxx uptake. For this to be more useful, there should be some discussion of what is causing what. The lack of attitudinal variables makes such a discussion tough.

I support publication.

Reviewer #2:

Remarks to the Author:

This is a very good and very important paper. Among the exceptional strengths is that there is no sampling error. The study essentially includes the entire population of Finland. There is variability in vaccination status - because up about 24% of the population had remained unvaccinated through October of 2021. In addition to health status information, the data set includes extensive demographic and health information and the investigators had access to individual level genetic information on the entire population.

For these and other reasons, reporting this information will be of great value. I have just a few minor concerns that should be noted in the discussion. These include:

1. The analysis considered 2890 predictor variables. Of course, with this unusually large number of predictors, there is always a danger of capitalizing on chance relationships. The discussion section might address this issue in more detail.
2. Along similar lines, it might be valuable to comment on the robustness of the Lasso procedure. Although the methodology is widely used, it might not be familiar to all readers of Nature Human Behavior.
3. This is a very minor point, but the paper includes several sentences that are hard to follow because they use double negatives. For example, "Among individuals with labor income, those in the lowest income decile had a significantly higher chance of not uptaking vaccination compared to individuals in the 40%-50% income decile bin...." I think it would be easier to say, "... were

significantly less likely to take the vaccine compared to....”

4. The evidence from the study shows that those with mental health diagnosis were less likely to take the vaccine. It is my understanding that the prevalence of some diagnosis, such as schizophrenia, are higher in Finland than in most other countries. If so, a comment on this issue may be valuable.

5. Does the observation that Memantine, was associated with a higher non-vaccination rate suggest that there is an association between Alzheimer's disease and acceptance of vaccination? Or, does this simply mean that those with cognitive impairment have relinquished the decision making process to others?

6. It may be helpful to briefly discuss the underlying construct associated with low income, poor education, substance abuse, etc. Presumably, these are not independent factors. Instead, they may represent a core factor of social disconnection. We know from lots of social epidemiology studies that these factors are associated with poor health outcomes.

7. Further, it is difficult to disentangle the social versus genetic contributions to these factors

8. Overall, the conclusions are clearly supported by the evidence. However, I was less clear on how the results would support national pro vaccination campaigns.

In summary, this is a well executed study evaluating an important problem using a very unique data resource. It will make a valuable contribution to the research literature.

Reviewer #3:

Remarks to the Author:

This paper investigates COVID19 vaccine uptake in a large Finnish population of 3 million. This is important in the contemporary context of the ongoing COVID-19 situation, may have generalizability/importance to other situations (eg influenza)?

Using nationwide health registers provide a massive sample size and the potential to discover risk factors. Inclusion of genetics to assess trait correlations and potential causal inference are of interest.

How were 2,890 predictors confined to 12 'consistent' categories? I liked the 12 categories broken out and explained, please add at least an additional sentence to "Selection and definition of the phenotypic predictors" to justify or explain the logic applied for the manually curated categories. Why not present the total model of all predictors at the top of Figure 2a? Why age and sex as a comparator, is there an expectation of prediction from these two variables? Might be better to label it as a null comparator given the low AUC. Might be useful to also include an a prior known predictor comparator to better highlight what is gained in this analysis, especially considering that everything other than long-term care exceeds the value of age and sex?

I particularly like the deep dive into the 99th percentile to identify factors associated with vaccination. Somethings seems peculiar or I fear I have misunderstood something in Figure 3b; the impact on unvaccinated vs vaccinated appears to be symmetrical? Is this the case, is this expected? The GWAS and rG are interesting, but in this case MR could be particularly enlightening. I see this was done in one direction (Severity -> vaccination). Why not test the reverse direction? It also wasn't clear (apologies to the authors if I missed it) how the instruments were selected? Were the

SNPs genomewide significant? If not, might they try a lower threshold to increase power ($5e-05$)? Further, why not include all 3 primary traits from the (critically ill, hospitalized, reported infection). Related, but it wasn't obvious which 2 of the 3 traits were used. 'Severity' could be critically ill or hospitalized, susceptibility seems likely to be reported infection? Answers to some of these might clarify whether there is a 'lack of causal relationship' or if there isn't power to discover this relationship.

This work is interesting, important and timely. I consider my comments minor. Clarifications on some of the methods (particularly for the genetics) would help this reviewers (and future readers) understanding.

Reviewer #4:

The XGB and Lasso models were applied to answer the specific clinical hypotheses. I summarize my major comments below:

- Overall it is not clear the experimental procedure employed by the authors for evaluating the performance of the XGB model and the most discriminative predictors. Although the authors claim (row 193 and row 755): "We studied the importance of the 12 categories of predictors in predicting COVID-19 vaccination uptake using machine learning models (XGBoost [17]) trained separately for 195 each category in a randomly sampled 80% of the study population and evaluated on the remaining 20%." it is not clear how the splitting procedure is implemented. Does the splitting procedure is stratified across outcomes?

- Why do the authors perform downsampling to speed up the training process of XGB and Lasso? The authors should provide more evidence that this process does not lead to loose relevant information for the training phase. Moreover, it is not clear if the downsampling procedure also helped to deal with the naturally imbalanced setting of this task.

- The authors set balanced class weighting as one of the hyperparameters of the XGB model. They should provide more evidence on how this hyperparameter may help to deal with the imbalanced setting

- Why do the authors use the SHAP method to extract the interpretability of the XGB model? The authors should better highlight the advantage (i.e local interpretability) of using this agnostic SHAP methodology rather than tree-based criteria.

- What is the rationale behind the use of a logistic regression model for evaluating the predictive power of each feature?

- How do the authors evaluate the statistical significance of the computed metrics (e.g AUC, feature importance)?

Author Rebuttal to Initial comments

Reviewer #1

Remarks to the Author: This is a fascinating paper using great data. Some minor comments.

Response: We thank the reviewer for their interest in our work and for their constructive feedback.

Comment 1. It seems like you are saying that the genetic indicators are both strongly and weakly related. Some of the discussion comes of as inconsistent.

Response: We thank the reviewer for the comment and agree that the distinction between a robust association, i.e. a significant effect passing genome-wide Bonferroni correction ($p < 5 \times 10^{-8}$), and the effect size needed to be clarified within the discussion. We have now added at page 10 wording prior to discussing the PGS results which hopefully clarifies such a distinction.

As is the case for many complex traits, the significant SNPs identified are robustly associated, but the effect sizes were small in isolation. Summing these effects into a PGS produced a weak, but not insignificant, predictor of COVID-19 vaccination.

Comment 2. The criticisms of the survey works are unnecessary. Maybe you want to point to specific papers instead of making these blanket criticisms on pages 81-95. The good thing about surveys is that many of them, particularly for COVID vaxx choices, have been fairly comprehensive. And, I am not sure that your data fully overcomes the sampling bias problems.

Response: After re-reading our summary of survey works of predictors of COVID-19 vaccination status, we agree with the reviewer that some of the limitations of earlier work came out unnecessarily harsh. We think that nation-wide data does overcome the sampling bias issue, since essentially the whole population is included in the study. However, we have now reconsidered the wording regarding some limitations of earlier studies with regard to reporting bias at page 3:

Third, surveys ~~are affected by reporting bias, either voluntary or involuntary, and can collect only a limited set of information limiting the power of epidemiological and machine learning analyses~~

And added text at page 2 pointing out that surveys usually include many attitudinal variables that are not available in registries and thus these two different types of studies are better seen as complementing each other

While offering important information about self-perceived reasons for vaccine hesitancy, and including

information that are not available in nation-wide registries, studies based on survey data have several limitations.

Comment 3. One concern I have is that many of your predictor variables are themselves outcomes, no different than COVID vaxx uptake. For this to be more useful, there should be some discussion of what is causing what. The lack of attitudinal variables makes such a discussion tough.

Response: We agree with the reviewer that the study setup we have used is not assessing whether/which of the registry-based predictors are causal to the vaccination uptake. We are also not making such claims, but rather report associations between the predictors (measured before the start of the pandemic) and the vaccination uptake and concentrate on how well vaccination uptake in general can be predicted based on registry and genetic information.

Investigating possible causal relationships between the registry-based predictors and vaccination uptake is further complicated by the overlapping nature of many of the variables and data sources, as demonstrated in Figure 4 of the manuscript. Finding causal factors underlying vaccine hesitancy is of course an interesting research question, and could be studied for example with causal inference methods.

We have now extended the use of Mendelian Randomization to assess whether certain biological traits are causally related to vaccination uptake in the revised version of the manuscript (see **updated Supplementary Table 9**). Note that this examination was limited to traits with strong biological evidence supporting the SNPs (height, type 2 diabetes, BMI) selected as the instrument variables to satisfy the assumptions of the Mendelian Randomization technique. We have described the results of these new analysis on page 8:

Similarly, no causal relationship of height or type 2 diabetes on vaccination uptake was found. However, higher Body Mass Index (BMI) was causally related to decreased vaccination uptake ($MR_{IVW}=0.121$, $SE=0.033$, $p\text{-value}=2.1\times 10^{-4}$), with no evidence of unbalanced pleiotropy ($MR_{\text{eggerintercept}}=-1.3\times 10^{-3}$, $SE=1.5\times 10^{-3}$, $p\text{-value}=0.39$).

We acknowledge that due to the nature of the registry data, we did not have access to attitudinal variables typically addressed in surveys that might ultimately capture some of the underlying reasons for not vaccinating. We suggest at page 9 that some of these attitudinal aspects might be latent in the data and could be uncovered by leveraging information across multiple variables.

Future work using dimensionality reduction techniques to obtain latent factors might help to reduce information redundancy and to identify some of underlying attitudinal aspects that are not directly

measured in the registers.

Reviewer 2

Remarks to the Author: This is a very good and very important paper. Among the exceptional strengths is that there is no sampling error. The study essentially includes the entire population of Finland. There is variability in vaccination status - because up to about 24% of the population had remained unvaccinated through October of 2021. In addition to health status information, the data set includes extensive demographic and health information and the investigators had access to individual level genetic information on the entire population.

For these and other reasons, reporting this information will be of great value. I have just a few minor concerns that should be noted in the discussion. These include:

Answer: We thank the reviewer for the positive comments. We wish to clarify, however, that the individual level genetic information was on a subset of the Finnish population. We used release 9 of the FinnGen dataset (N=273,765). Given the potential for confusion, we have clarified the following sentence at page 7 relating to the genetic analysis confirming that we do not have such data available within the entire population.

We therefore performed a genome-wide association study of COVID-19 vaccination uptake in FinnGen (N=273,615; ~8.6% of FinRegistry participants aged 30-80) and the Estonian Biobank (N=145,615), restricted to European ancestry.

Comment 1. The analysis considered 2890 predictor variables. Of course, with this unusually large number of predictors, there is always a danger of capitalizing on chance relationships. The discussion section might address this issue in more detail.

Response: We thank the reviewer for asking for clarification regarding this important point. When considering this large number of predictors it is possible that some associations are significant by chance (false positives), and/or that high correlation between some of the variables makes the interpretation of the results difficult.

Regarding the first issue, we corrected the p-values of the logistic regression models for individual predictors with the Benjamini-Hochberg method. To clarify this, we have added a sentence to the main text of the revised manuscript at page 4 stating that the reported p-values from the logistic regression models have been adjusted for multiple hypothesis testing:

Benjamini-Hochberg method was used to adjust the p-values for multiple hypothesis testing.

Regarding the issue of correlations between the predictors, we carried out extensive analyses by removing each of the variable categories at a time to obtain a picture of how much of the predictive information is shared between the different predictor categories (see **Figure 4**). In theory, dimensionality-reduction approaches could be used to identify independent components explaining vaccination uptake. We have now mentioned this possibility in the Discussion section of the main text at page 9.

Future work using dimensionality reduction techniques to obtain latent factors might help to reduce information redundancy and to identify some of underlying attitudinal aspects that are not directly measured in the registers.

Comment 2. Along similar lines, it might be valuable to comment on the robustness of the Lasso procedure. Although the methodology is widely used, it might not be familiar to all readers of Nature Human Behavior.

Response: It is a good suggestion that Lasso could have been better introduced already in the submitted version of the manuscript. We have now added a sentence describing Lasso and added a reference to (Tibshirani, 1996) to the main text of the revised version of the manuscript at page 4:

Lasso is a logistic regression model penalized with L1 norm that acts both as a regularizer and a feature selector.

Comment 3. This is a very minor point, but the paper includes several sentences that are hard to follow because they use double negatives. For example, “Among individuals with labor income, those in the lowest income decile had a significantly higher chance of not uptaking vaccination compared to individuals in the 40%-50% income decile bin...” I think it would be easier to say, “.... were significantly less likely to take the vaccine compared to....”

Response: We thank the reviewer for raising this important point about readability of the text. We have now removed double negatives, and otherwise tried to streamline some sentences in the revised version of the manuscript at page 5:

Among individuals with labor income, those in the lowest income decile were significantly less likely to take the vaccine compared to individuals in the 40%-50% income decile bin (OR=1.08, 95% CI: 1.08-1.09; Figure 2c).

At page 5:

Memantine (Other anti-dementia drugs), a medication used to treat symptoms of cognitive impairment such as Alzheimer's disease, was associated with lower vaccination rate (OR 1.04 and 95% CI 1.03-1.05).

At page 5:

We found that having an unvaccinated mother increases the risk of not vaccinating (OR=1.31, 95% CI:1.31-1.32) more than having an unvaccinated father (OR=1.23, 95% CI: 1.22-1.23) or having any unvaccinated siblings (OR=1.17, 95% CI: 1.16-1.17).

At page 6:

In the test set, the top 1% of individuals with the lowest predicted probability to vaccinate (N=6,385) had an observed vaccination rate of only 18.8% vs 90.3% when considering everyone in the test set (Figure 3a).

Comment 4. The evidence from the study shows that those with mental health diagnosis were less likely to take the vaccine. It is my understanding that the prevalence of some diagnosis, such as schizophrenia, are higher in Finland than in most other countries. If so, a comment on this issue may be valuable.

Response: We thank the reviewer for raising this point about generalization of the results, which certainly apply to the mental health diagnoses as you suggested, but also potentially to other variables that can have differing prevalence across countries. We have mentioned this limitation at page 10:

First, generalizability outside Finland and to non-European ancestries is unclear and replication in other countries is needed to understand the generalizability of our findings across different populations. Generalizability can be impacted by differences in disease prevalence (e.g. schizophrenia being more common in Finland than in most European countries²⁹), but also because of varying methods of organizing national vaccination programs across vulnerable groups.

Comment 5. Does the observation that Memantine, was associated with a higher non-vaccination rate suggest that there is an association between Alzheimer's disease and acceptance of vaccination? Or, does this simply mean that those with cognitive impairment have relinquished the decision making process to others?

Response: Both of these explanations appear plausible, but we would like to avoid making

causal claims based on this observational data alone. Observationally, people with Alzheimer's disease have, on average, lower education than individuals without Alzheimer's, which might partially account for the observed association in our study (Given that education and socioeconomic variables were clearly associated with vaccination uptake). Furthermore, the association between memantine purchases and vaccination uptake (OR=1.04) was not very large. However, an important implication of this association would be, similarly to the effect of mental health and substance abuse diagnoses, to ensure that those people that are experiencing cognitive decline are ensured equal access to vaccination.

Comment 6. It may be helpful to briefly discuss the underlying construct associated with low income, poor education, substance abuse, etc. Presumably, these are not independent factors. Instead, they may represent a core factor of social disconnection. We know from lots of social epidemiology studies that these factors are associated with poor health outcomes.

Response: The reviewer raises an important point - many of the predictors associated with lower COVID-19 vaccination uptake, such as low income, low education or substance abuse, are likely not independent, but different manifestations of a “core factor of social disconnection”. We feel it is important to notice that there are many correlating measures in large registries and electronic health records. For example we found that predictions made using drug purchase history, and income are surprisingly correlated. We have expanded the discussion of independence of the individual predictors and potential future directions in the main text of the revised manuscript at page 9:

Future work using dimensionality reduction techniques to obtain latent factors might help to reduce information redundancy and to identify some of underlying attitudinal aspects that are not directly measured in the registers.

Comment 7. Further, it is difficult to disentangle the social versus genetic contributions to these factors

Response: The reviewer notes an important point here. While we can establish which traits are genetically correlated with vaccination uptake, it cannot be determined based on the analyses presented here whether the correlations arise due to purely biological reasons or due to shared socioeconomic factors underlying both of the correlated traits. In fact, we believe it is the latter for most of the associations.

Comment 8. Overall, the conclusions are clearly supported by the evidence. However, I was less clear on how the results would support national pro vaccination campaigns.

Response: We feel that one of the key conclusions of the study is that we were able to identify groups of people with higher risk to not vaccinate (such as people with mental health or substance abuse diagnoses and medications). This information could be used to try to target or tailor information campaigns on vaccination programs, to make vaccination centers more accessible to the identified risk groups or to even target financial incentives for vaccination. This information could also be used to target future research efforts into elucidating what drives lower vaccination rates in these groups. We acknowledge the reviewer's point that these conclusions could have been more clearly presented in the submitted version of the manuscript. To address this, we have expanded the final paragraph of the discussion section of the main text in the revised version of the manuscript at page 11 to better communicate these points.

Using a machine learning approach, we could identify groups of individuals with significantly lower risk to vaccinate than the average population. These results provide potential avenues for targeted interventions supporting COVID-19 and possibly other national immunization programs, for example by designing vaccine distribution centers that are more accessible to individuals less likely to vaccinate. In addition, a recent study has shown that financial incentives for vaccination do not have negative unintended consequences [REF]. Targeting financial incentives based on vaccination probabilities could be a cost-effective measure to increase effectiveness of immunization programs.

Comment 9: In summary, this is a well executed study evaluating an important problem using a very unique data resource. It will make a valuable contribution to the research literature.

Response: We thank the reviewer for their interest in our work and for their constructive feedback.

Reviewer 3:

Remarks to the Author: This paper investigates COVID19 vaccine uptake in a large Finnish population of 3 million. This is important in the contemporary context of the ongoing COVID-19 situation, may have generalizability/importance to other situations (eg influenza)? Using nationwide health registers provide a massive sample size and the potential to discover risk factors. Inclusion of genetics to assess trait correlations and potential causal inference are of interest.

Response: We thank the reviewer for their interest in our work. Indeed as the reviewer points out, we feel that the approach presented here could in future be applied also to other public health issues.

Comment 1: How were 2,890 predictors confined to 12 ‘consistent’ categories? I liked the 12 categories broken out and explained, please add at least an additional sentence to “Selection and definition of the phenotypic predictors” to justify or explain the logic applied for the manually curated categories.

Response: We thank the reviewer for pointing out that the logic of how the 12 predictor categories were defined could have been better explained in the original submitted manuscript. The logic of the predictor categories derives from both expert knowledge and from the different original source registers. Each of the source registers includes variables that are thematically similar. For example, the Finnish Register of Social Assistance includes variables that describe when, why and how long individuals have received social benefits, and thus predictors created from this register were assigned to the Social benefits predictor category. We have now further clarified this logic in the Methods section at page 13 of the revised manuscript following the suggestion by the reviewer

Division into the 12 predictor categories was based on expert knowledge and largely reflects the source registers of the predictors - the FinRegistry dataset is aggregated from several individual thematic registers. For example, the Social benefits category predictors come from the Finnish Register of Social Assistance that is a collection of variables detailing received periods of income support.

Comment 2: Why not present the total model of all predictors at the top of Figure 2a? Why age and sex as a comparator, is there an expectation of prediction from these two variables? Might be better to label it as a null comparator given the low AUC. Might be useful to also include an a priori known predictor comparator to better highlight what is gained in this analysis, especially considering that everything other than long-term care exceeds the value of age and sex?

Response: We agree with the reviewer that including the AUC of the full model already in Figure 2a helps in putting the performances of the individual-category models into perspective. We have now included the AUC of the full XGBoost model as a dashed vertical line in revised Figure 2a and marked the baseline/null predictor with a dotted vertical line

(see also below).

Figure 2. *a)* Area under receiver-operator characteristics curve (AUC) for XGBoost classifiers trained using predictors from different predictor categories (each model also includes the baseline predictors age and sex). Error bars show 95% confidence intervals computed using bootstrapping. Number of predictors within each category is indicated on top of the corresponding bar. The black vertical dashed line indicates the performance of the full XGBoost model using all predictors. The black dotted vertical line corresponds to an XGBoost model allowed to use only age and sex as predictors. Most of the predictor categories perform better than this baseline model, with Income and Drug purchases being the most predictive categories. *b)* AUC from Lasso classifiers trained separately for each of the individual predictors (models also include the baseline predictors age and sex), grouped by the categories. Some of the interesting highly predictive predictors have been highlighted (for a fully annotated list of AUCs of individual predictors, see Supplementary Table 2). *c)* Association between labor income in 2019 and COVID-19 vaccination uptake. Odds ratio from a logistic regression model using income percentile bins as predictors and adjusting for age and sex. The 40%-50% percentile bin was used as a reference category. Error bars indicate 95% confidence intervals for odds ratios computed using bootstrapping. *d)* Associations between previous disease diagnoses and COVID-19 vaccination status. Odds ratio from a logistic regression model using a binary disease indicator as predictor and adjusting for age and sex. Some of the interesting predictors are highlighted. Predictors with multiple hypothesis testing-adjusted p-value > 0.01 (Benjamini-Hochberg method), and prevalence among vaccinated <1000 are not shown. Error bars indicate 95% confidence intervals for odds ratios computed using bootstrapping. For a fully annotated list of ORs of individual predictors, see Supplementary Table 3.

As the reviewer notes, age and sex should be used as a “null comparator”. We agree, and this is exactly how they were used already in the submitted version of the manuscript, termed as the “baseline model”. This could have been more clearly written in the submitted version of the manuscript, and we have now clarified this point in the revised version of the manuscript at page 4:

As many of the predictors highly correlate with age and sex, comparison to the performance of the

baseline model shows how much additional predictive information each category contains.

Age and sex were also selected as the baseline comparator due to lack of earlier registry-based predictors of COVID-19 vaccination uptake. It is worth noting that **Figure 4** shows that many different registry-based prediction models with similar prediction performances to the main prediction model can be derived based on register data, depending on what type of data is available.

Comment 3: I particularly like the deep dive into the 99th percentile to identify factors associated with vaccination. Somethings seems peculiar or I fear I have misunderstood something in Figure 3b; the impact on unvaccinated vs vaccinated appears to be symmetrical? Is this the case, is this expected?

Response: We thank the reviewer for raising this issue. The reviewer has understood correctly that the average impact of the features on the XGBoost model output presented separately for vaccinated and unvaccinated in **Figure 3b** of the submitted manuscript was misleading. As we are dealing with binary classification, the impact measured with |SHAP| is by construction symmetrical with respect to the classes. We have updated Figure 3b in the revised manuscript by removing the stratification by vaccinated vs unvaccinated, and reported average impact on vaccination status in general instead (see also below).

Figure 3. a) Fractions of unvaccinated individuals in the test set as a function of centile bins of predicted probabilities to not vaccinate from the full XGBoost model. The 99th centile bin comprises 6,385 individuals that have only an 18.8% (95% CI: 17.9%-19.8%) chance of vaccinating. The error bars indicate 95% confidence intervals computed using bootstrapping. The black dashed line indicates the average fraction of unvaccinated individuals in the study population. **b)** Mean absolute SHAP values²⁰ computed for all individual predictors used in the full XGBoost model. Higher values indicate higher average impact of the predictor. The top 20 most important predictors are shown for clarity. The error bars indicate 95% confidence intervals computed using

bootstrapping (some uncertainty estimates are very small).

Comment 4: The GWAS and rG are interesting, but in this case MR could be particularly enlightening. I see this was done in one direction (Severity -> vaccination). Why not test the reverse direction?

Response: We chose not to test the reverse direction (vaccination → severity) as we were not confident that the underlying genetic signal for vaccination uptake is biological in origin and more likely reflects gene-environment correlation. The assumptions behind Mendelian Randomization are therefore more difficult to justify, so we opted not to test this direction. This contrasts with the COVID-19 phenotypes which have produced sensible biological pathways from gene-analyses in the original GWAS. Further, as we were limited to using the FinnGen only summary statistics for vaccination uptake (to remove sample overlap), we could only use one significant SNP, limiting our power to test for a causal effect.

Comment 5: It also wasn't clear (apologies to the authors if I missed it) how the instruments were selected? Were the SNPs genomewide significant? If not, might they try a lower threshold to increase power (5e-05)?

Response: We thank the reviewer for highlighting this lack of clarity. We have now expanded the methods section to be more explicit in our chosen parameters for the mendelian randomization. We have also included a sensitivity analysis where we lower the threshold to 5e-5 as suggested (results are included in updated Supplementary Table 9). The results remained consistent at this threshold where we continue to find no evidence of a causal impact of COVID-19 phenotypes on vaccination uptake. We added text at page 21:

For all tests, instrumental variables were selected using a p-value threshold of 5×10^{-8} and clumped using the default parameters (window size=10,000kb; $r^2 < 0.001$). As a sensitivity analysis, we repeated the MR for COVID-19 phenotypes using a less stringent p-value threshold of 5×10^{-5} .

Comment 6: Further, why not include all 3 primary traits from the (critically ill, hospitalized, reported infection). Related, but it wasn't obvious which 2 of the 3 traits were used. 'Severity' could be critically ill or hospitalized, susceptibility seems likely to be reported infection? Answers to some of these might clarify whether there is a 'lack of causal relationship' or if there isn't power to discover this relationship.

Response: In addition to the severity phenotype which originally corresponded to the

hospitalization phenotype, we have expanded all PGS and MR analyses to cover the full breadth of COVID-19 phenotypes, namely - critical illness, hospitalization and susceptibility. We added text in the methods at page 21:

PGSs for COVID-19 resulting in critical illness, resulting in hospitalization, and susceptibility were calculated using the same method, but association with COVID-19 vaccination uptake was performed in the full sample due to the lack of sample overlap. Release 7 of the COVID-19 Host Genetic Initiative with FinnGen and 23andMe excluded (COVID-19 critical illness: $N_{\text{cases}} = 17,962$, $N_{\text{controls}} = 867,353$; COVID-19 hospitalization: $N_{\text{cases}} = 44,549$, $N_{\text{controls}} = 2,018,071$; COVID susceptibility: $N_{\text{cases}} = 155,026$, $N_{\text{controls}} = 2,445,292$) were used as summary statistics to calculate PGS¹⁵.

To test the causal effect of COVID-19 critical illness, hospitalization and susceptibility severity on vaccination status, we used Mendelian Randomization⁴². MRBase was used to run two sample mendelian randomization⁴³. For the exposures, we selected the same summary statistics used for the PGS analysis¹⁵ whereas for the outcome, we selected the summary statistics for vaccination status from the FinnGen sample only as to prevent sample overlap.

We added text in the results at page 8:

To test if individuals at higher genetic risk for COVID-19 critical illness, hospitalization and susceptibility were more or less likely to vaccinate, we built PGS for each of the three COVID-19 phenotypes using Release 7 from the COVID-19 host Genetic initiative¹⁵, which includes mostly studies collected before the start of the vaccination campaigns. Individuals with higher PGS for each COVID-19 phenotype were less likely to receive the vaccine. However, the association was modest (critical illness: OR = 1.02, 95%CI: 1.01 - 1.04, hospitalization: OR = 1.04, 95%CI: 1.02 - 1.05; susceptibility: OR = 1.02, 95%CI: 1.01 - 1.04 per 1 standard deviation in PGS) partially due to the PGS for COVID-19 being a weak predictor of COVID-19.

Mendelian randomization analysis indicates a lack of causal relationship between COVID-19 phenotypes and vaccination uptake (Supplementary Table 9).

Moreover, based on the above suggestion, we decided to expand the MR analyzes and include three traits - height, BMI and type 2 diabetes - which show evidence of genetic overlap and likely have a clear biological origin. This resulted in a finding that higher BMI is causally associated with decreased vaccination uptake. We added text in the Methods section on page 21:

We additionally tested for the causal effect of Height, Type 2 Diabetes and BMI as the traits either had significant genetic correlations or the genome-wide significant SNPs indicated overlapping effects. As sample overlap was not an issue for these traits, the meta-analyzed summary statistics for vaccination status were used to improve power.

We added text in the results at page 8:

Similarly, no causal relationship of height or type 2 diabetes on vaccination uptake was found. However, higher BMI was causally related to decreased vaccination uptake ($MR_{ivw}=0.121$, $SE=0.033$, $p\text{-value}=2.1\times 10^{-4}$), with no evidence of unbalanced pleiotropy ($MR_{eggerintercept}=-1.3\times 10^{-3}$, $SE=1.5\times 10^{-3}$, $p\text{-value}=0.39$).

Comment 7: This work is interesting, important and timely. I consider my comments minor. Clarifications on some of the methods (particularly for the genetics) would help this reviewers (and future readers) understanding.

Response: We thank the reviewer for their comments and suggestions. We feel that addressing the above mentioned issues has strengthened the manuscript.

Reviewer 4

The XGB and Lasso models were applied to answer the specific clinical hypotheses. I summarize my major comments below:

Comment 1: Overall it is not clear the experimental procedure employed by the authors for evaluating the performance of the XGB model and the most discriminative predictors. Although the authors claim (row 193 and row 755): “We studied the importance of the 12 categories of predictors in predicting COVID-19 vaccination uptake using machine learning models (XGBoost [17]) trained separately for 195 each category in a randomly sampled 80% of the study population and evaluated on the remaining 20%. “ it is not clear how the splitting procedure is implemented. Does the splitting procedure is stratified across outcomes?

Response: We thank the reviewer for pointing out this issue and apologize that the evaluation of the XGBoost model performance was not explained in more detail in the submitted manuscript. The sampling was implemented by taking all the cases, and sampling completely at random five times as many controls. This was explained in the Methods of the submitted manuscript, but not in the main text. We have now explained how the sampling was done also in the main text at page 4 of the revised manuscript.

Training was conducted in a randomly sampled 80% of the study population and evaluated in the

remaining 20%. To speed up training, controls were downsampled in the training data so that five randomly sampled controls (vaccinated) were included for each case (unvaccinated).

Comment 2: Why do the authors perform downsampling to speed up the training process of XGB and Lasso? The authors should provide more evidence that this process does not lead to loose relevant information for the training phase. Moreover, it is not clear if the downsampling procedure also helped to deal with the naturally imbalanced setting of this task.

Response: We thank the reviewer for pointing out that further justification for the performed downsampling might be beneficial. Ultimately, downsampling was required for two reasons.

1) Training the 4,097 Lasso models for Figure 4b would have required an unfeasible amount of time without downsampling the controls, and 2) We did not have access to computing resources with enough memory to train a full XGBoost model with ~3,000 variables and ~2.6M training samples.

We did perform experiments to check that the downsampling does not significantly affect the prediction performance and apologize that these were not reported already in the submitted version of the manuscript. Results comparing the AUCs of models trained with the full and the downsampled datasets are now reported in a new **Extended Data Figure Figure 2a** and commented in the main text at page 4 of the revised manuscript (see also the figure below).

To speed up training, controls were downsampled in the training data so that five randomly sampled controls (vaccinated) were included for each case (unvaccinated). Downsampling did not significantly affect XGBoost model predictions (Extended Data Figure 2a).

Extended Data Figure 2. a) Downsampling controls does not negatively affect the machine learning model predictions. AUCs for models trained with all cases and five randomly sampled controls per each case (orange) and for models trained with the full training data without downsampling (black). Predictors used by the models are indicated on the x-axis. All AUCs correspond to XGBoost models, except for the Full model (indicated with blue color), where the AUCs were computed for the Lasso model, as the full XGBoost model could not be trained without downsampling the controls due to memory issues. **b)** Class weighting has a negligible effect on the XGBoost model predictions. AUCs for XGBoost models trained with balanced class weighting (orange) versus with no class weights (blue). In both cases, five controls per each case were sampled randomly for the training data.

Predictors used by the models are indicated on the x-axis. In both panels, the error bars indicate 95% confidence intervals computed using bootstrapping. All models include the baseline predictors age and sex.

It is worth noting, that all the reported results both in the submitted and the revised manuscript were evaluated in the full test set, i.e. without any downsampling.

Comment 3: The authors set balanced class weighting as one of the hyperparameters of the XGB model. They should provide more evidence on how this hyperparameter may help to deal with the imbalanced setting

Response: We thank the reviewer for pointing out that this hyperparameter was not explored in the submitted version of the manuscript. We now provide a comparison of the AUCs of the XGBoost models used in this study trained with and without balanced class weighting in a new **Extended Data Figure 2b**. We observe a small but significant improvement in AUC when not using balanced class weights for many of the XGBoost models. Due to the minor effect, we have decided to keep the main analyses as is, and report these results in the **new Extended Data Figure 2b** (see also figure below). We also added some text about this aspect to the Methods section of the revised manuscript at page 18:

Balanced class weighting was used to penalize misclassification of both vaccinated and unvaccinated equally. Classification performance was marginally better without using balanced class weighting (Extended Data Figure 2b).

Extended Data Figure 2. a) Downsampling controls does not negatively affect the machine learning model predictions. AUCs for models trained with all cases and five randomly sampled controls per each case (orange) and for models trained with the full training data without downsampling (black). Predictors used by the models are indicated on the x-axis. All AUCs correspond to XGBoost models, except for the Full model (indicated with blue color), where the AUCs were computed for the Lasso model, as the full XGBoost model could not be trained without downsampling the controls due to memory issues. **b)** Class weighting has a negligible effect on the XGBoost model predictions. AUCs for XGBoost models trained with balanced class weighting (orange) versus with no class weights (blue). In both cases, five controls per each case were sampled randomly for the training data.

Predictors used by the models are indicated on the x-axis. In both panels, the error bars indicate 95% confidence intervals computed using bootstrapping. All models include the baseline predictors age and sex.

Comment 4: Why do the authors use the SHAP method to extract the interpretability of the XGB model? The authors should better highlight the advantage (i.e local interpretability) of using this agnostic SHAP methodology rather than tree-based criteria.

Response: As the reviewer points out, there exist different methods for interpreting feature importances of XGBoost models. We decided to employ SHAP to extract feature importances from our XGBoost model as some of the more traditional feature importance metrics used for tree-based models have undesirable properties. For example, Gini impurity summarizes how often a feature is used in different nodes of the tree, which easily leads to wrong conclusions of relative feature importance in models such as ours that mixes binary and continuous variables (as continuous variables can be split more times than binary ones). SHAP expresses feature importance as Shapley values, which have a theoretical justification from game theory. SHAP values are computed for each sample, and they express the average contribution of a feature value to the prediction. When averaged over a large set of samples, like we have done in **Figure 3b**, SHAP values give the average contribution in the specific set of samples. Given these properties and the widespread use, we believe that SHAP values are a valuable metric to interpret feature importance.

Comment 5: What is the rationale behind the use of a logistic regression model for evaluating the predictive power of each feature?

Response: Logistic regression (with the L1 penalty) was used to evaluate the predictive power of the individual features. The rationale was to establish how predictive were the individual features alone without interactions, and then to evaluate the increase in prediction

that could be achieved by leveraging combinations of predictors in the same model using XGBoost. This rationale was not adequately presented in the submitted version of the manuscript, as pointed out by the reviewer, but has now been introduced in the main text of the revised manuscript at page 4:

The rationale was to establish a baseline that can be achieved using individual predictors.

Comment 6: How do the authors evaluate the statistical significance of the computed metrics (e.g AUC, feature importance)?

Response: Statistical significance of the AUCs were estimated by computing 95% confidence intervals using bootstrapping, as explained in the Methods section and figure captions. We apologize that this was not clearly stated in the main text of the submitted manuscript. We have now added a sentence to the main text of the revised manuscript at page 4 indicating how uncertainty of the AUCs was estimated.

Confidence intervals for AUC were computed using bootstrapping (see Methods)

We also thank the reviewer for pointing out that the uncertainty of the |SHAP| values were not reported in the submitted version of the manuscript. We have now added 95% confidence intervals of the |SHAP| values to the revised Figure 3b, which was also modified to report mean |SHAP| values without dividing the effects for the different classes following the comment number 3 from Reviewer 3 (see also figure below). We also added the following sentence to the Methods at page 18:

Confidence intervals for mean absolute SHAP values were computed by bootstrapping the test set 2000 times.

Figure 3. a) Fractions of unvaccinated individuals in the test set as a function of centile bins of predicted probabilities to not vaccinate from the full XGBoost model. The 99th centile bin comprises 6,385 individuals that have only an 18.8% (95% CI: 17.9%-19.8%) chance of vaccinating. The error bars indicate 95% confidence intervals computed using bootstrapping. The black dashed line indicates the average fraction of unvaccinated individuals in the study population. **b)** Mean absolute SHAP values [19] computed for all individual predictors used in the full XGBoost model. Higher values indicate higher average impact of the predictor. The top 20 most important predictors are shown for clarity. The error bars indicate 95% confidence intervals computed using bootstrapping (some uncertainty estimates are very small).

Decision Letter, first revision:

22nd February 2023

Dear Dr. Ganna,

Thank you for submitting your revised manuscript "Nationwide health, socioeconomic and genetic predictors of COVID-19 vaccination uptake" (NATHUMBEHAV-22112922A). It has now been seen by the original referees and their comments are below. As you can see, the reviewers find that the paper has improved in revision.

We will therefore be happy in principle to publish it in Nature Human Behaviour, pending minor revisions to comply with our editorial guidelines. Congratulations.

We are now performing detailed checks on your paper and will send you a checklist detailing our editorial and formatting requirements within a week. Please do not upload the final materials and make any revisions until you receive this additional information from us.

Sincerely,

Arunas Radzvilavicius, PhD
Editor, Nature Human Behaviour
Nature Research

Reviewer #1 (Remarks to the Author):

I only identified a few concerns in the first round. In this round, my concerns have been addressed, and in rereading the revised draft, I have no further comments. I support publication.

Reviewer #2 (Remarks to the Author):

The authors have been very responsive to my earlier comments. In addition, I looked at some of the changes in response to other reviewers. Overall, they did a good job.

This is a good paper and I have no remaining concerns

Reviewer #3 (Remarks to the Author):

I'm mostly satisfied with reviewer responses, particularly as I originally stated my comments as minor. I would have liked to see reverse direction MR with vaccination uptake but find the power limitation a very reasonable consideration against doing so.

Reviewer #4 (Remarks to the Author):

The authors answered my methodological concerns. In particular, I appreciated the further experiments the authors performed to justify downsampling and class weights. As a minor concern, I suggest that the authors should better highlight how the employment of class weights did not lead to a significant gain, mainly due to the fact that they already performed downsampling.

Final Decision Letter:

Dear Dr Ganna,

We are pleased to inform you that your Article "Nationwide health, socioeconomic, and genetic predictors of COVID-19 vaccination status in Finland", has now been accepted for publication in Nature Human Behaviour.

Please note that *Nature Human Behaviour* is a Transformative Journal (TJ). Authors whose manuscript was submitted on or after January 1st, 2021, may publish their research with us through the traditional subscription access route or make their paper immediately open access through payment of an article-processing charge (APC). Authors will not be required to make a final decision about access to their article until it has been accepted. IMPORTANT NOTE: Articles submitted before January 1st, 2021, are not eligible for Open Access publication. Find out more about Transformative Journals

With best regards,

Arunas Radzvilavicius, PhD
Senior Editor, Nature Human Behaviour
Nature Research